# Lessons Learned: A Multi-Agent Framework for Code LLMs to Learn and Improve

**Yuanzhe Liu**[1][*]  **Ryan Deng**[2][†]  **Tim Kaler**[2]  **Xuhao Chen**[2,3]
**Charles E. Leiserson**[2]  **Yao Ma**[1]  **Jie Chen**[4]

[1]Rensselaer Polytechnic Institute    [2]Massachusetts Institute of Technology
[3]Michigan State University    [4]MIT-IBM Watson AI Lab, IBM Research
{liuy72,may13}@rpi.edu   {ryandeng,tfk,cxh,cel}@mit.edu
chenjie@us.ibm.com

## Abstract

Recent studies show that LLMs possess different skills and specialize in different tasks. In fact, we observe that their varied performance occur in several levels of granularity. For example, in the code optimization task, code LLMs excel at different optimization categories and no one dominates others. This observation prompts the question of how one leverages multiple LLM agents to solve a coding problem without knowing their complementary strengths a priori. We argue that a team of agents can learn from each other's successes and failures so as to improve their own performance. Thus, a lesson is the knowledge produced by an agent and passed on to other agents in the collective solution process. We propose a lesson-based collaboration framework, design the lesson solicitation–banking–selection mechanism, and demonstrate that a team of small LLMs with lessons learned can outperform a much larger LLM and other multi-LLM collaboration methods.

## 1 Introduction

Code optimization refers to rewriting the code such that it performs the same task more efficiently [32]. The efficiency is predominantly measured by the runtime, but it could also refer to storage, energy, or other resource consumptions. Code optimization is a less explored coding task in the context of AI compared with the popularly studied code generation task [62], but it is a natural, and sometimes key, step in the software development cycle. Code performance can be improved in many ways, such as using a lower-complexity algorithm, caching and memorization, data alignment, vectorization, and parallelization [50, 5]. Many of these approaches require low-level system knowledge that code LLMs are not explicitly trained with.

We are interested in exploring the use of multiple LLM-powered agents for code optimization. We use multiple LLMs because no one LLM performs the best on all problems, even if they are trained with extensive code data. The common practice of benchmarking uses aggregated performance to rank models [6], but the best-ranked model may not be the best performer for every problem. We take the ParEval benchmark [39], which consists of programming problems in scientific computing, for example. When three open-source, small models and two GPT models are evaluated on this benchmark (see Appendix A for details), GPT-4o [43] is the overall winner. However, on the "geometry" category of problems, Qwen7B [25] outperforms GPT-4o by $2.5\times$ in terms of speedup; and on the "histogram" category, Deepseek7B [20] and Qwen14B [25] outperform GPT-4o by $1.6\times$ (Figure 5). Additionally, among the three open-source models, Qwen7B excels at "search," while

---

[*]Part of the work was done while YL visited MIT-IBM Watson AI Lab, IBM Research.

[†]Part of the work was done while RD interned at MIT-IBM Watson AI Lab, IBM Research.
 Code implementation is available at https://github.com/MITIBM-FastCoder/LessonL.

**Original code**

```
for (int i = 0; i < n; ++i)
  for (int j = 0; j < n; ++j)
    for (int k = 0; k < n; ++k)
      C[i][j] += A[i][k] * B[k][j];
```

Naive implementation of matrix multiplication $C = AB$.

**Improved code, round 1**

```
for (int i = 0; i < n; ++i)
  for (int k = 0; k < n; ++k)
    for (int j = 0; j < n; ++j)
      C[i][j] += A[i][k] * B[k][j];
```

**Lesson:** Reordering loops improves cache locality and increases performance. The order of (i,k,j) out of 6 different permutations often performs the best, because of how caches work.

**Improved code, round 2**

```
#pragma omp parallel for
for (int i = 0; i < n; ++i)
  for (int k = 0; k < n; ++k)
    for (int j = 0; j < n; ++j)
      C[i][j] += A[i][k] * B[k][j];
```

**Lesson:** Using OpenMP to parallelize the for-loop further improves performance. Parallelizing only the outermost loop performs the best, due to sufficient parallelism and less parallel scheduling overhead.

Figure 1: Successively improving matrix-matrix multiplication in C with lessons.

Deepseek7B excels at "scan" and "dense_la" and Qwen14B excels at "sparse_la," "reduce," "fft," and "sort" (Table 5). Such varied performance suggests the opportunity of exploiting the complementary strengths of different LLMs to deliver the best solution.

How does one make use of multiple agents to solve a coding problem? In this work, we advocate the concept of *lessons* inspired by classroom experience. A student's problem-solving skills are not only taught by a teacher at class but also developed through peer learning. For example, after receiving the graded homework, Student A, who cannot complete the solution to a problem may consult with Student B, who earns a perfect score, on tips of the correct steps toward the solution. Meanwhile, Student A can also benefit from learning from Student C, who comes up with a wrong solution, to avoid making similar mistakes. Each student learns from the success and failure lessons of others to improve their own problem-solving skills. LLM agents behave similarly. Pre-training is analogous to classroom teaching, while a pre-trained LLM can improve its skills through prompted with lessons.

A lesson means any information that helps an LLM better solve the problem at hand. For code optimization, such lessons may be optimization strategies applicable to the current code, common pitfalls that programmers are trapped in, or performance feedback from profilers. Note that a code's performance can be improved by a combined use of multiple strategies, step by step. Figure 1 shows the initial steps of a classic example of engineering the performance of matrix-matrix multiplications, which can be improved by 3000x in speedup under extensive optimization [50]. The applied strategies shown are loop reordering and parallelization of for-loops. In a multi-LLM setting, such strategies, or lessons, can be iteratively summarized by one or a few LLMs and learned by others, so that they collectively improve the code performance.

In this work, we propose the framework **LessonL** (pronouced as "lesson-nell") for multiple LLM agents to collaboratively solve problems. Central to the framework is the lesson mechanism for agents to improve their collective intelligence through learning from each other. The main technical innovation is a solicitation–banking–selection framework that generates, deposits, and filters lessons incurred during the collective problem-solving process. Although code optimization is the focal application of LessonL, we demonstrate its use for other tasks (code generation) as well.

LessonL is a novel multi-agent framework that resembles how humans learn to solve problems. Compared with other collaboration frameworks, such as where agents play different roles in the solution process [49, 12, 23, 27, 46], or where agents independently propose solutions that are subsequently communicated and aggregated [52, 14, 28, 37, 60] (see Section 2 for literature review), our framework has a few advantages. First, agents do not need to be distinguished by pre-specified roles, as their complementary strengths for a particular problem may be unknown a priori. Second, communication and prompt contents are economic since lessons are more concise than codes. Third,

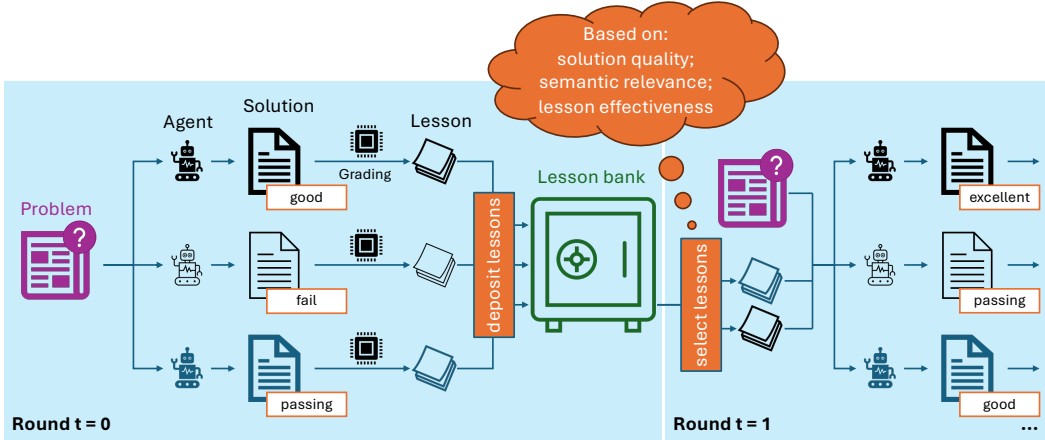

Figure 2: The LessonL framework (which may repeat multiple rounds).

more importantly, lessons are interpretable and reusable, allowing the explication of coding knowledge and the creation of educational materials.

Our work contributes the following:

1. a finding that LLMs have complementary strengths even on a fine level of tasks (Appendix A);

2. a novel lesson-based framework for multiple agents to collectively solve problems (Section 3);

3. state-of-the-art performance on code optimization and code generation benchmarks (Section 4.2);

4. empirical evidence that a team of small LLMs can significantly outperform a much larger LLM under similar resource consumptions (Section 4.4);

5. representative code examples and lessons (Appendices K–M).

## 2 Related Work

**Multi-agent collaboration.** Recent advances in prompting techniques have enhanced LLMs' reasoning and planning capabilities for complex tasks like mathematics and coding. CoT [54], Self-Consistency [53], ReAct [58], and ToT [57] demonstrate improved problem-solving skills with reasoning while Reflexion [49] leverages self-generated feedback stored in episodic memory to enhance decision-making. As such, LLMs can be used as autonomous agents and recent research efforts propose ways for multiple agents to collaboratively solve problems. Popular multi-agent frameworks either place agents in different roles (such as planner, coder, debugger, reviewer, and tester in a software project) or have agents independently propose solutions, which are collectively refined and consolidated. For role-based methods, see AgentVerse [12], MetaGPT [23], MapCoder [27], Chat-Dev [46], Self-collaboration [13], SoA [26], AgentCoder [24], Agent Hospital [34], and DEI [61]. For individual solution proposals, see MoA [52], LLM-Debate [14, 15], LLM-Blender [28], DyLAN [37], and AgentPrune [60]. Our work LessonL belongs to the latter category. A unique contribution of LessonL is the lesson mechanism for agents to learn from each other and the explication of coding knowledge as a result of the collaborative learning.

**Code optimization.** It is an important, but under-explored, use case of LLM for code. Prior work focuses on specialized models such as HPC-Coder [40, 41] and HPC-Coder-V2 [9] for high-performance computing, all requiring curating and/or generating code data and fine-tuning. Besides the usual fine-tuning approaches, PIE [51] proposes additional adaptation techniques, including retrieval-based prompting, performance-conditioning, and self-play. Among the few agentic approaches, SBLLM [17] retrieves optimization examples from an external dataset and Self-Refine [38] iteratively refines the code based on self-generated feedback. In contrast, our multi-agent framework allows the use of more than one agent and it does not rely on an external code dataset.

For extended discussions and more related work, see Appendix B.

# 3 The LessonL Framework

Our multi-agent collaboration framework is centered around *lessons*, which can be any knowledge or information that helps an agent better solve the problem at hand. When a team of agents participates in the collective problem-solving process, such knowledge is solicited through inspecting each agent's solution and is deposited into a bank for other agents to access. Hence, the solution process becomes iterative, looping between using the existing lessons to update the solutions and using the updated solutions to generate new lessons. For code optimization, such an iterative process can progressively improve the code performance. This process is pictorially illustrated in Figure 2 and sketched in Algorithm 1:

---

**Algorithm 1** The LessonL Framework (Sketched)

---

1: Each agent generates an initial solution and the lesson for it. Deposit lessons to the bank.
2: **for** round $t = 1, \ldots, T$ **do**
3:     Select $k$ lessons from the bank.
4:     Based on the selected lessons, each agent generates an updated solution.
5:     Each agent generates a new lesson for the updated solution. Deposit lessons to the bank.
6:     Adjust the effectiveness of the $k$ selected lessons.
7: **end for**
8: Return the best solution.

---

In what follows, we elaborate on a few key components of this framework. The full algorithm is given in Appendix C. We also discuss its extension to other coding tasks, such as code generation.

## 3.1 Lesson Solicitation

Every solution comes with a lesson, either positive or negative. Such lessons explain why the solution is correct or what makes it wrong. For code optimization, multiple tools can be used to grade the output code, such as the compiler, the clock, test cases, and profilers. Consider four resulting scenarios:

(a) speed up,    (b) slow down,    (c) functional incorrectness,    (d) syntax error.

We solicit lessons by referring the original code and the modified code, A and B, respectively, and prompting the agent with the resulting scenario and asking for explanations. For (a) and (b), we supplement the scenario description with the measured speedup and for (d), we include the error message reported by the compiler. Such information helps the agent reason with the code pair and articulate precise lessons. For (c), we do not include the test cases because LLMs tend to reason specifically about the test cases, resulting in insufficiently general lessons.

The detailed prompt templates are given in Appendix D.

## 3.2 Lesson Banking and Selection

In each round, $n$ lessons are generated and deposited to the bank, one from each agent. Then, after several rounds, there accumulate many lessons. It is unwise to feed all of them to the prompt when asking the agent to improve the code, because they may exceed the prompt capacity and also the token consumption can become too costly. Hence, the framework is run with at most $k$ lessons in each round. In practice, it suffices to set $k$ to be, or slightly larger than, $n$.

A set of lesson selection criteria is in place. First, naturally we want to use lessons about high speedups, because they point to the right directions of optimization. However, positive lessons alone cannot address certain limitations of LLMs, such as the lack of guarantee on code correctness. Hence, negative lessons are still valuable, because they can help the agents avoid similar mistakes. It is, however, challenging to decide which negative lessons are more important than others. Finally, we also consider relevance, in case an agent hallucinates irrelevant lessons. For this, we treat the original code as the query and retrieve semantically relevant lessons from the bank, in a manner similar to retrieval augmented generation [33].

Algorithmically, the selection mechanism goes as follows. If the bank has no more than $k$ lessons, pick them all. Otherwise, sort the lessons according to speedup (more on this in the next subsection)

and pick the top $\lceil k/2 \rceil$. Then, sort the remaining lessons according to their cosine similarity with the original code and pick the top $\lfloor k/2 \rfloor$. Any powerful embedding models for code can be used to compute the cosine similarity; for example, CodeBERT [16].

### 3.3 Effectiveness Adjustment

The effectiveness of a lesson $z$ is naturally measured by the speedup $s$ — the higher the speedup, the more useful the lesson. Note that $s$ was calculated when $z$ was created together with code $y$. In other words, $s$ is the speedup of code $y$ over the original code. One's view on the effectiveness of $z$ may change when this lesson is selected to apply later. For example, if the lesson later yields code $y'$ with a worse speedup $s' < s$, should we keep selecting it only because the original $s$ is great?

To more effectively select high-speedup lessons, we need some adjustment to the speedup dynamically. Rather than sorting them according to $s$, we introduce an adjustment factor $f$ and sort the lessons according to $s \times f$ instead, where $f$ accounts for the performance of the lesson $z$ when actually applied. Specifically, when $z$ is applied in some round $t$, it incurs speedups $s_j^{(t)}$ for each agent $j$. We initialize a correction variable $c$ with zero and add $1 + \epsilon$ to it whenever $s_j^{(t)} > s$, or add $1 - \epsilon$ when $s_j^{(t)} < s$, for some $\epsilon \in (0, 1)$. After looping over all $n$ agents, the adjustment factor $f$ is defined as $c/n$. A value greater than 1 means more output codes enjoy a speedup $> s$ by applying the lesson.

### 3.4 Extension to Other Coding Tasks

The LessonL framework is general. It can be extended to other coding tasks provided that the key components discussed above are properly adapted. For example, in Python code generation, lessons are needed to be solicited for only one scenario: functional incorrectness, because if the output code passes all test cases, the iteration immediately terminates. We give the prompt templates in Appendix E for completeness. Additionally, we will still perform lesson banking, but $k$ lessons are selected based on the number of test cases passed and the semantic relevance instead of speedup.

## 4 Experiments

We perform a comprehensive set of experiments to evaluate LessonL. The main finding is that LessonL enables an effective collaboration among LLMs to perform code optimization and other coding tasks. Using the LessonL framework, a team of small LLMs that collaborate through the sharing of learned lessons can outperform larger LLMs and multi-LLM collaboration methods.

### 4.1 Setup

**Benchmarks.** We use six coding benchmarks to perform the evaluation. (1) **ParEval** [39] includes 60 coding tasks (per programming mode) related to scientific and parallel computing. We experiment with the serial and OpenMP modes, as they are less demanding on the computational resources required for evaluation. ParEval was originally designed for code generation (i.e., write the code given verbal description). We adapt it to code optimization (i.e., write a faster version of a given source code); see the adaptation details in Appendix F. (2) **PolyBench** [45] contains 30 numerical tasks with static control flows from domains like linear algebra, image processing, physics, and statistics, adapted for code optimization by us. (3) **HumanEval** [11] consists of 164 programming problems, assessing language comprehension, algorithms, and basic mathematics. (4) **HumanEval+** [36] extends HumanEval with 80x more test cases. (5) **MBPP** [4] consists of around 1,000 crowd-sourced entry-level Python problems covering fundamentals and standard library use. (6) **MBPP+** [36] extends MBPP with 35x more test cases. The last four benchmarks evaluate code generation.

**LLMs.** We use seven models in total. Three are open-source, small models: **Deepseek7B** (deepseek-coder-7b-instruct-v1.5) [20], **Qwen7B** (Qwen2.5-Coder-7B-Instruct) [25], and **Qwen14B** (Qwen2.5-Coder-14B-Instruct) [25]; two are GPT models not trained with reasoning data: **GPT-4o mini** (gpt-4o-mini) [42] and **GPT-4o** (gpt-4o) [43]; and two are reasoning models: **DeepseekR1-14B** (DeepSeek-R1-Distill-Qwen-14B) [19] and **OpenAI o3** (o3) [44]. The open-source models are used to evaluate multi-agent collaborations; GPT-4o mini is closed-source but its size is comparable to the open-source models; while GPT-4o is much larger and it sets a strong baseline for single-agent

performance. Additionally, DeepSeekR1-14B and OpenAI o3 are two recent reasoning models that make use of the test-time scaling approach to achieve strong reasoning capabilities.

**Baselines.** We compare LessonL with four categories of models/methods. (1) **Single-agent standard prompting**. All the above non-reasoning LLMs are experimented with. Among the open-source models, preliminary findings suggest that Qwen14B slightly outperforms the other two (see Table 5 in Appendix A); hence, we omit the results of these two in subsequent tables to save space. (2) **Single-agent trained with reasoning data**. Both DeepSeekR1-14B and OpenAI o3 are off-the-shelf models equipped with reasoning capabilities. (3) **Single-agent reasoning or reflection**. We use Qwen14B as the agent and experiment with CoT [54] and Reflexion [49]. CoT applies a chain-style thought process to reason about the steps, while Reflexion iteratively reflects on the task feedback to improve the solution. (4) **Multi-agent collaboration**. We experiment with MapCoder [27] and MoA [52]. MapCoder uses agents for example retrieval, solution planning, coding, and debugging. For our purpose, all agents are Qwen14B. In contrast, each agent in MoA independently codes the solution and refines the aggregated solution. We use the open-source models as agents and use GPT-4o as the aggregator. Similarly, **in our framework LessonL**, the agents are open-source models.

For details on baseline implementation, hyperparameters, hardware, and timing, see Appendix G.

## 4.2 Benchmark Results

We evaluated the performance of LessonL on code optimization and code generation tasks. For code optimization, we studied the ability of LessonL to optimize serial and parallel code drawn from the ParEval and PolyBench benchmarks. For code generation, we investigated LessonL's performance on HumanEval, HumanEval+, MBPP, and MBPP+.

**Code optimization task.** In this task, models are given a correct program and are tasked with generating a faster version of that code while maintaining correctness evaluated by using test cases. The **speedup** achieved by a model is measured as the ratio of the runtime between the original and the new code. We evaluate the performance of a model by measuring the geometric mean speedup achieved over the set of codes in the benchmark. The geometric mean is preferred over the arithmetic mean because it is more resilient to large outliers that can cause a single code to unduly influence the average. For example, when using the arithmetic mean an algorithmic optimization that improves the asymptotic runtime of a code from $\Theta(n^2)$ to $\Theta(n \log n)$ could result in a $1000\times$ speedup for sufficiently large input, and result in the arithmetic average becoming a poor measure for a model's ability to optimize diverse sets of codes. In the case where a new code is incorrect or slower than the original, we consider the speedup to be $1$. Similar to [51], which uses the same convention, our rationale is any new code may be discarded in favor of keeping the original code.

**Code optimization results.** Table 1 presents the results. We compare the performance of LessonL with three single-agent models (Qwen14B, GPT-4o-mini, and GPT-4o), two reasoning models (DeepseekR1-14B and OpenAI o3), and four prompting / agentic methods (CoT, Reflexion, MapCoder, and MoA). We report each method's average speedup, proportion of correct code, and the proportion of codes that achieve a speedup greater than $2\times$.

A few observations follow. First, LessonL achieves either the best or the second best results across metrics and benchmarks (with OpenAI o3 taking the other position generally). This highlights the attractiveness of LessonL's novel collaboration mechanism, which enables agents to learn form each other and iteratively improve solutions. Second, multi-agent collaborations generally surpass single-agent approaches (except using OpenAI o3 which comes with native reasoning). MapCoder and MoA, which use role specialization or solution aggregation, are often second to LessonL. Third, achieving large speedups is more challenging for serial code than for OpenMP code. In serial mode, most methods yield an speedup below $2\times$, with less than 20% of solutions surpassing $2\times$ speedup. Conversely, in OpenMP mode, most methods achieve more than $2\times$ speedup, with more than 40% of solutions surpassing $2\times$. This is expected, as many scientific computing and image processing tasks are inherently parallelizable. For instance, LessonL achieves an average $3.4\times$ speedup with 8 threads across both benchmarks.

We additionally report the improvement of LessonL over the single-agent models in Appendix J.1, Table 10. As motivated earlier, these models have complementary (but unknown a priori) strengths. LessonL improves over them on a majority of the problem categories (some highly significant), while experiences a minor degradation on the remaining categories due to randomness in LLM generation.

Table 1: Comparison of model/method performance for two code optimization benchmarks. "Correct" means correctness; "> 2×" means proportion of problems achieving speedup > 2×; "Speedup" is the geometric mean speedup across all problems in the benchmark. The results are reported as the mean and standard deviation over three runs.

| **ParEval** | Serial mode | | | OpenMP mode | | |
|---|---|---|---|---|---|---|
| | Correct | > 2× | Speedup | Correct | > 2× | Speedup |
| Qwen14B | 0.67 ± 0.03 | 0.14 ± 0.01 | 1.60 ± 0.03 | 0.67 ± 0.01 | 0.48 ± 0.00 | 2.28 ± 0.02 |
| GPT-4o mini | 0.77 ± 0.03 | 0.14 ± 0.01 | 1.57 ± 0.12 | 0.70 ± 0.03 | 0.55 ± 0.04 | 2.72 ± 0.09 |
| GPT-4o | 0.80 ± 0.00 | 0.16 ± 0.03 | 1.72 ± 0.11 | 0.73 ± 0.05 | 0.58 ± 0.05 | 2.93 ± 0.30 |
| DeepseekR1-14B | 0.56 ± 0.05 | 0.12 ± 0.05 | 1.51 ± 0.19 | 0.44 ± 0.03 | 0.31 ± 0.03 | 1.68 ± 0.07 |
| OpenAI o3 | 0.77 ± 0.02 | **0.23 ± 0.04** | **2.21 ± 0.16** | 0.72 ± 0.03 | 0.58 ± 0.03 | **3.55 ± 0.27** |
| CoT | 0.61 ± 0.03 | 0.12 ± 0.03 | 1.57 ± 0.12 | 0.65 ± 0.04 | 0.51 ± 0.03 | 2.31 ± 0.03 |
| Reflexion | 0.67 ± 0.03 | 0.18 ± 0.00 | 1.88 ± 0.20 | 0.64 ± 0.01 | 0.46 ± 0.02 | 2.27 ± 0.06 |
| MapCoder | 0.88 ± 0.02 | 0.15 ± 0.02 | 1.85 ± 0.08 | 0.83 ± 0.05 | 0.58 ± 0.02 | 3.43 ± 0.17 |
| MoA | 0.73 ± 0.04 | 0.16 ± 0.02 | 1.76 ± 0.08 | 0.74 ± 0.04 | 0.59 ± 0.02 | 2.77 ± 0.11 |
| LessonL | **0.91 ± 0.02** | 0.21 ± 0.01 | 2.16 ± 0.11 | **0.86 ± 0.01** | **0.62 ± 0.02** | 3.46 ± 0.03 |

| **PolyBench** | Serial mode | | | OpenMP mode | | |
|---|---|---|---|---|---|---|
| | Correct | > 2× | Speedup | Correct | > 2× | Speedup |
| Qwen14B | 0.46 ± 0.12 | 0.02 ± 0.02 | 1.03 ± 0.04 | 0.49 ± 0.02 | 0.46 ± 0.02 | 2.21 ± 0.06 |
| GPT-4o mini | 0.41 ± 0.08 | 0.06 ± 0.02 | 1.10 ± 0.03 | 0.55 ± 0.07 | 0.48 ± 0.05 | 2.42 ± 0.31 |
| GPT-4o | 0.51 ± 0.02 | 0.06 ± 0.02 | 1.12 ± 0.03 | 0.60 ± 0.12 | 0.55 ± 0.04 | 2.73 ± 0.21 |
| DeepseekR1-14B | 0.18 ± 0.04 | 0.05 ± 0.04 | 1.08 ± 0.07 | 0.30 ± 0.03 | 0.25 ± 0.04 | 1.68 ± 0.09 |
| OpenAI o3 | 0.64 ± 0.02 | **0.28 ± 0.02** | **1.71 ± 0.21** | **0.72 ± 0.04** | 0.56 ± 0.02 | 3.15 ± 0.23 |
| CoT | 0.51 ± 0.03 | 0.02 ± 0.04 | 1.06 ± 0.04 | 0.39 ± 0.03 | 0.37 ± 0.04 | 1.92 ± 0.16 |
| Reflexion | 0.41 ± 0.02 | 0.08 ± 0.07 | 1.12 ± 0.07 | 0.56 ± 0.23 | 0.52 ± 0.04 | 2.59 ± 0.22 |
| MapCoder | 0.64 ± 0.02 | 0.12 ± 0.04 | 1.17 ± 0.03 | 0.55 ± 0.04 | 0.44 ± 0.05 | 2.23 ± 0.18 |
| MoA | 0.32 ± 0.07 | 0.08 ± 0.02 | 1.12 ± 0.04 | 0.61 ± 0.07 | 0.54 ± 0.05 | 2.70 ± 0.34 |
| LessonL | **0.77 ± 0.04** | 0.13 ± 0.04 | 1.32 ± 0.09 | 0.71 ± 0.03 | **0.62 ± 0.02** | **3.40 ± 0.12** |

Table 2: Comparison of model/method performance (pass@1) for four code generation benchmarks. GPT-4o and GPT-4o mini results are taken from the EvalPlus leaderboard [36].

| | **HumanEval** | **HumanEval+** | **MBPP** | **MBPP+** |
|---|---|---|---|---|
| Qwen14B | 0.866 | 0.817 | 0.831 | 0.741 |
| GPT-4o mini | 0.884 | 0.835 | 0.854 | 0.722 |
| GPT-4o | 0.884 | 0.872 | 0.876 | 0.722 |
| CoT | 0.884 | 0.848 | 0.854 | 0.741 |
| Reflexion | 0.902 | 0.841 | 0.889 | 0.765 |
| MapCoder | 0.890 | 0.823 | 0.862 | 0.679 |
| MoA | 0.909 | 0.872 | 0.897 | **0.770** |
| LessonL | **0.915** | **0.878** | **0.899** | 0.765 |

In the same appendix section, we show an example (Problem 40 of ParEval) where an agent critically relies on the lessons from other agents to successfully solve the problem.

**Code generation.** While code optimization is the main testbed of LessonL, we also experiment with a widely studied coding task—code generation. Table 2 compares the results for four commonly used benchmarks by using the correctness (pass@1) metric. We see that LessonL is the best in three out of four benchmarks, with MoA coming next. In the remaining benchmark, the ranking positions of these two frameworks are swapped. This observation echos that of code optimization, corroborating the usefulness of multi-agent collaboration for solving coding tasks and highlighting the effectiveness of our lesson framework.

## 4.3 Ablation Study

We conducted additional studies on LessonL to investigate the impact of its design components, the number of collaboration rounds, and the number of agents.

Table 3: Ablation study. Benchmark: ParEval (serial and OpenMP modes). The results are reported as the mean and standard deviation over three runs.

| | | Serial mode | | | OpenMP mode | | |
|---|---|---|---|---|---|---|---|
| | | Correct | $> 2\times$ | Speedup | Correct | $> 2\times$ | Speedup |
| (0) | LessonL | $0.91 \pm 0.02$ | $\mathbf{0.21 \pm 0.01}$ | $\mathbf{2.16 \pm 0.11}$ | $\mathbf{0.86 \pm 0.01}$ | $\mathbf{0.62 \pm 0.02}$ | $3.46 \pm 0.03$ |
| (1) | $k$ high-speedup only | $\mathbf{0.92 \pm 0.02}$ | $\mathbf{0.21 \pm 0.01}$ | $2.08 \pm 0.04$ | $0.83 \pm 0.02$ | $0.58 \pm 0.00$ | $3.20 \pm 0.04$ |
| (2) | $k$ high-relevance only | $0.91 \pm 0.02$ | $0.18 \pm 0.02$ | $1.96 \pm 0.08$ | $0.84 \pm 0.02$ | $0.61 \pm 0.01$ | $3.40 \pm 0.16$ |
| (3) | No speedup adjustment | $\mathbf{0.92 \pm 0.03}$ | $\mathbf{0.21 \pm 0.03}$ | $2.05 \pm 0.05$ | $\mathbf{0.86 \pm 0.02}$ | $0.61 \pm 0.02$ | $3.28 \pm 0.10$ |
| (4) | Random $k$ lessons | $0.91 \pm 0.02$ | $0.19 \pm 0.02$ | $2.03 \pm 0.05$ | $0.82 \pm 0.02$ | $0.61 \pm 0.11$ | $\mathbf{3.47 \pm 0.28}$ |
| (5) | No lessons | $0.89 \pm 0.01$ | $0.20 \pm 0.00$ | $2.05 \pm 0.01$ | $0.82 \pm 0.01$ | $0.56 \pm 0.01$ | $3.01 \pm 0.16$ |

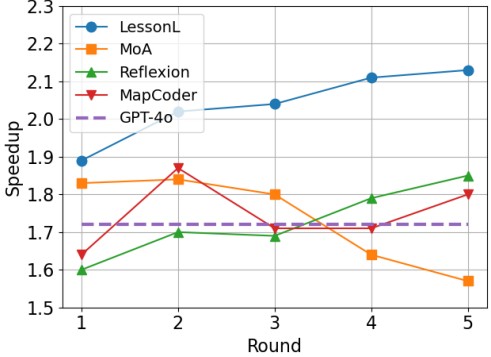

Figure 3: Performance over rounds (or called "layers"). Benchmark: ParEval (serial mode).

Table 4: Performance comparison across different numbers of agents. Benchmark: ParEval (serial mode).

| #Agents | Correct | $> 2\times$ | Speedup |
|---|---|---|---|
| 1 | 0.67 | 0.14 | 1.60 |
| 3 | 0.90 | 0.23 | 2.12 |
| 4 | 0.93 | 0.22 | 2.14 |
| 5 | 0.93 | 0.20 | 2.09 |
| 6 | 0.95 | 0.23 | 2.21 |

**Ablating components of the framework.** We analyzed the lesson selection mechanisms. The results are illustrated in Table 3 and compare five variations of the LessonL framework: (0) the full version of LessonL; (1) lessons selected based only on speedup; (2) lessons selected based only on relevance; (3) lessons selected based on speedup and relevance, but without a speedup adjustment; (4) random selection of lessons; (5) no lessons used.

Table 3 reports the results. We see that most ablated variants suffer a decrease of speedup. In serial mode, variants without high-speedup lesson prioritization (2,4) suffer most, while in OpenMP mode, variants (1,3) show larger performance drops. This trend also applies to the proportion of problems with >2x speedup. This highlights different optimal strategies for each mode. Serial mode benefits from high-speedup lesson selection with dynamic adjustments, while OpenMP mode favors high-relevance lessons and speedup adjustments. Interestingly, in serial mode, the correctness metric sometimes benefits from ablations, even though the relationship between lessons and correctness is indecisive. While variant (4) marginally outperforms LessonL in OpenMP speedup (+0.01), LessonL demonstrates significantly better stability (0.03 vs 0.28 standard deviation). Finally, The consistent underperformance of variant (5) across all metrics confirms that lessons are fundamental to LessonL.

**Varying the number of iterations.** Figure 3 plots the performance (speedup) achieved by LessonL and alternative methods when varying the number of collaboration rounds. The concept of a "round" is different across methods: for MoA, a round is a "layer", and for MapCoder, it is a "debugging round". The performance of GPT-4o is also included to provide a baseline for comparison. While LessonL and Reflexion consistently benefit from using more rounds (see Appendix J.2, Table 11, where we push the number of rounds for LessonL to 10), the same is not true for MoA and MapCoder. The performance of MoA actually decreases when increasing the number of rounds, and there is no clear trend when using more than two rounds in MapCoder. Even when using a much smaller model, Reflexion surpasses GPT-4o and continues to improve when further increasing the number of rounds. In contrast, MoA's performance drops below GPT-4o when using more than three rounds, even though MoA actually uses the GPT-4o model as an aggregator.

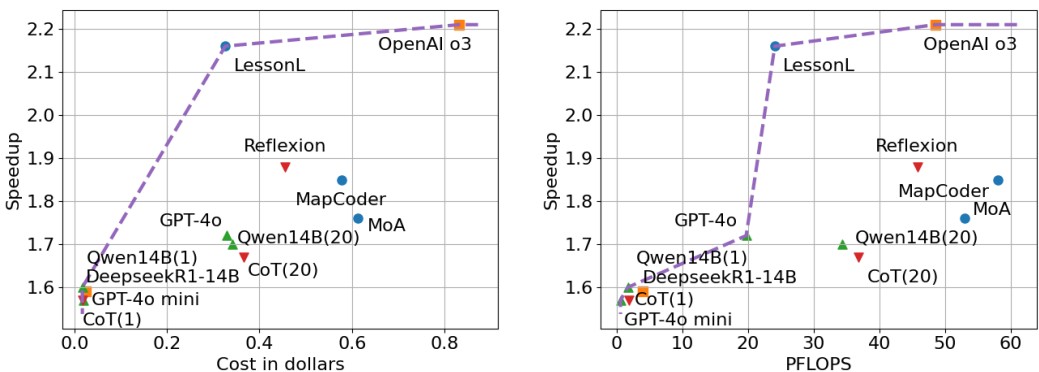

Figure 4: Performance versus costs and latency. Benchmark: ParEval (serial mode). The dashed line is the Pareto front.

We studied the outputs of each method and provide examples in Appendix H. For a program that finds the $k$th smallest element of an array, we observe that MoA implements the divide-and-conquer algorithm in early rounds, achieving high speedup. In subsequent rounds, however, MoA introduces overheads and eventually resorts to a simple solution that calls std::nth_element, which results in slower code. In contrast, in the example of performing convolutions on an image, LessonL increases the speedup step by step by removing redundant zero-additions, separating boundary and interior computations, and avoiding extra data copies. These examples illustrate how the lesson mechanism can selectively inject useful information to aid improvements in coding solutions.

**Varying the number of agents.** To see how LessonL performs in a larger scale, we added three more agents in order: Llama-3.1-8B-Instruct [18], Qwen2.5-7B-Instruct [48], and phi-4 (14B) [2]. These models have a similar size to the three already used. The results in Table 4 suggest that generally, using more agents lead to better performance. This observation is uniform for all metrics. However, the greatest improvement comes from using three agents; using more generates diminishing returns.

## 4.4   Cost Analysis

Code performance is not the only dimension that determines the effectiveness of a code model, considering the complex interplay of monetary costs, time costs and LLM calls in single-LLM and multi-LLM methods. We investigate where LessonL stands for the code optimization task by considering budget constraints with respect to money and time. For this, we follow [52] and extract the pricing information of LLMs from API providers or guesstimate the price based on similar models. We also use the number of floating point operations (FLOPS) as a proxy of latency when considering the time cost. See Appendix I for the detailed estimation. The estimation may be time sensitive (for example, price varies over time and across providers) and may include educated guesses (for example, the sizes of closed-source models), but it paints a reasonable picture of the performance landscape.

Figure 4 shows two speedup-versus-cost plots for ParEval, with the Pareto front indicated by the dashed line. Besides the afore-mentioned models/methods, we add Qwen14B(20) and CoT(20), which prompt the respective model/method 20 times and select the fastest code (an inference-time scaling approach [7]). We see a few clusters from the plots. First, LessonL is Pareto optimal because of its superior speedup and reasonable costs compared with competitors. Second, Qwen14B, DeepseekR1-14B, GPT-4o mini, and CoT are Pareto optimal, or nearly optimal, because of the low costs in both money and time. Third, OpenAI o3 is also Pareto optimal because of the superior speedup it achieves, despite at a much higher cost. We consider all these models/methods to be cost-effective. For the remaining methods, inference-time scaling (CoT(20) and Qwen14B(20)) does not help and neither do Reflexion or multi-agent methods. In fact, MapCoder and MoA appear to be the least cost-effective, because they not only iterate multiple times, but also use multiple agents. This signifies the challenge of multi-agent methods, which are generally the most costly, and in contrast reflects the attractiveness of LessonL. Finally, compared with the much larger model GPT-4o, LessonL with small models yields significantly better speedups by consuming similar resources.

### 4.5 Case Study

A few examples reveal the interesting lessons learned by the LLMs in our framework; see details in Appendices K (Geometry) and L (DFT). In the Geometry example, the original code finds the smallest distance among $n$ points in two dimensions by using the straightforward $O(n^2)$ implementation. The final optimized code uses a divide-and-conquer algorithm that reduces the complexity to $O(n \log n)$ and achieves a 74.31x speedup. This optimization is a complete change of algorithm and it is nontrivial even for human programmers. Several of the lessons affirm the benefit of divide-and-conquer while others point out syntactic errors and discourage the agents to repeat them.

More interesting is the DFT example, where the original code implements the straightforward algorithm in $O(N^2)$ cost. It is well known that DFT can be implemented by using the FFT algorithm in $O(N \log N)$ cost and some agents try to implement it, but they fail with subtle bugs. The lessons alert these failures and eventually the agents choose to optimize the code by precomputing the repeatedly used exponentials, yielding a considerable 10.83x speedup. Moreover, one lesson stands out by stating a syntactic error caused by using Intel intrinsic instructions. The evaluation pipeline and hardware do not support particular intrinsics and the involved agent later refrains from using them in the implementation.

There are also occasions when a problem is too challenging for the small LLMs to handle, even when they collaborate. We show an example in Appendix M (sum of prefix sum). The example involves various failures an LLM could produce: syntax errors, semantic errors, and lack of speedup. One mitigation is to integrate external knowledge to complement the self-reflected lessons.

## 5 Conclusions

We have presented a lesson-based framework LessonL for multiple agents to collaboratively solve coding problems. This framework allows each agent to learn from other agents' successes and failures to improve its own solution. Empirical evaluations indicate that with lessons learned over the course, our framework outperforms other methods to achieve state-of-the-art performance, and a team of small LLMs can significantly outperform a much larger LLM under a similar resource budget. Case studies show that the agents perform many skillful optimizations, such as performing divide-and-conquer and precomputation. An avenue of future research is to improve the autonomy of agents, including their decision-making ability in lesson selection.

**Limitations.** Even though experiment results demonstrate competitive latency of LessonL compared with large LLMs, learning from lessons will defer the time to first token, because lesson solicitation and subsequent trials are overheads. This may affect user experience. However, the negative impact may be offset by the rich and educational information—lessons—delivered together with the solution. Furthermore, LessonL has so far been applied only to function-level code snippets rather than repository-level tasks such as those in SWE-bench [29]. Extending LessonL to broader coding scenarios remains an important direction for future work.

**Broader impacts.** Code optimization is an important step of the software development cycle, especially for domains such as high performance computing, real-time animation, and transaction systems. This work develops a cost-effective approach to improving programmer productivity, by using LLMs to suggest optimized code. The interpretability of lessons lowers the barrier of mastering the low-level system knowledge required for optimization. Nevertheless, LLM outputs are not without flaws and practitioners should verify the lessons if they plan to apply the knowledge elsewhere.

## Acknowledgments and Disclosure of Funding

The research was supported by the National Science Foundation (NSF) under grant numbers NSF2406647 and NSF-2406648. This research was sponsored by the MIT-IBM Watson AI Lab and in part by the United States Air Force Research Laboratory and the United States Air Force Artificial Intelligence Accelerator and was accomplished under Cooperative Agreement Number FA8750-19-2-1000. The views and conclusions contained in this document are those of the authors and should not be interpreted as representing the official policies, either expressed or implied, of the United States Air Force or the U.S. Government. The U.S. Government is authorized to reproduce and distribute reprints for Government purposes notwithstanding any copyright notation herein.

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

# Appendix

# A    Complementary Strengths of LLM Agents

Section 1 motivates the use of multiple LLM agents to solve coding tasks, because it is unknown a priori which coding agent performs the best for each problem. We supplement the discussions with benchmarking results on the adapted ParEval benchmark [39] here (see Appendix F for the adaptation details). Figure 5 compares the overall performance of five LLMs, three open-source and two closed-source. Specifically, the models are:

- deepseek-coder-7b-instruct-v1.5, abbreviated as Deepseek7B;
- Qwen2.5-Coder-7B-Instruct, abbreviated as Qwen7B;
- Qwen2.5-Coder-14B-Instruct, abbreviated as Qwen14B;
- gpt-4o-mini, abbreviated as GPT-4o mini;
- gpt-4o, abbreviated as GPT-4o.

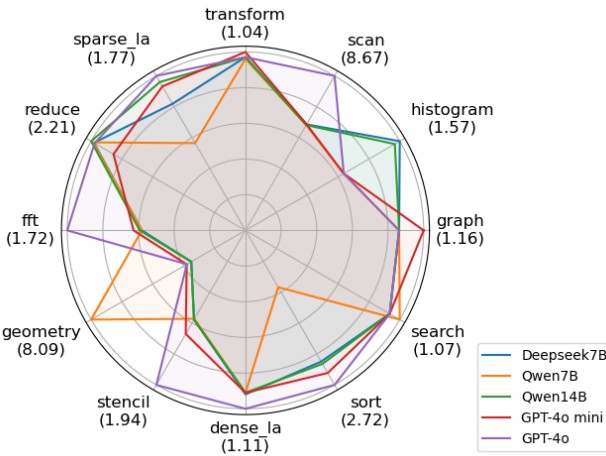

Figure 5: Comparison of models on their coding abilities for each category of the ParEval benchmark (serial mode). The coding ability is measured by the speedup when asked to optimize a code. Geometric mean of the speedups is taken over all problems in the same category. The number in the parenthesis is the maximum speedup for the corresponding category over all models. Performance for each category is normalized by the maximum speedup.

Moreover, in Table 5, we break down their performance into each category of the benchmark. We also highlight the best one among the open-source models. One sees that each model excels in some of the categories but no one dominates others.

Table 5: Speedup achieved by a model for each category of the ParEval benchmark (serial mode). The speedup is computed as the geometric mean. Three open-source models (i.e., excluding GPT-4o mini and GPT-4o) are compared with the best **boldfaced**.

|  | Deepseek7B | Qwen7B | Qwen14B | GPT-4o mini | GPT-4o |
|---|---|---|---|---|---|
| graph | **1.00** | **1.00** | **1.00** | 1.16 | 1.00 |
| histogram | **1.57** | 1.00 | 1.52 | 1.00 | 1.00 |
| scan | **5.95** | 5.91 | 5.91 | 5.95 | 8.67 |
| transform | **1.01** | 1.00 | **1.01** | 1.04 | 1.01 |
| sparse_la | 1.45 | 1.00 | **1.70** | 1.65 | 1.77 |
| reduce | 2.17 | 2.18 | **2.21** | 1.89 | 2.16 |
| fft | 1.00 | 1.00 | **1.02** | 1.08 | 1.72 |
| geometry | 2.84 | **8.09** | 2.84 | 3.08 | 3.07 |
| stencil | **1.12** | 1.11 | **1.12** | 1.30 | 1.94 |
| dense_la | **1.02** | 1.01 | 1.01 | 1.01 | 1.11 |
| sort | 2.31 | 1.00 | **2.35** | 2.51 | 2.72 |
| search | 1.00 | **1.07** | 1.00 | 1.00 | 1.00 |
| Overall | 1.57 | 1.50 | **1.59** | 1.58 | 1.78 |

# B  More Related Work

An LLM can be used as a problem solver in many ways. LATM [8] proposes a tooling framework wherein an LLM can create tools on its own and reuse them later. LATS [64] shows that LLMs can be integrated with Monte Carlo tree search for more effective planning. PHP [63] is an iterative framework where solutions are refined based on hints. L2MAC [22] introduces a system that mirrors the von Neumann architecture of a computer, utilizing a prompt program and file storage to generate long and coherent outputs. SFS [35] frames code generation as a black-box optimization problem, employing search techniques to enhance solution diversity and efficiency.

Moreover, multiple LLMs can collaborate for problem solving. MoA [52] orchestrates multiple LLM agents in a layered structure to aggregate and synthesize the solution. LLM-Debate [14, 15] enables structured inter-agent debates to reach consensus. LLM-Blender [28] ranks, filters and fuses top $k$ responses from multiple LLMs, improving the performance on instruction following. DyLAN [37] employs dynamic team adjustment for different tasks. AgentPrune [60] eliminates redundant communication through graph pruning. GPTSwarm [65] optimizes collaboration through graph-based interactions. ChatLLM [21] facilitates interactions among multiple dialogue-based agents, enabling them to provide feedback and collectively improve the decision-making process. EoT [59] promotes cross-model communication among LLMs, integrating various communication paradigms and a confidence evaluation mechanism to enhance reasoning accuracy. AutoGen [56] automates the development of agent-based applications, streamlining processes such as data collection, model training, and evaluation to efficiently adapt pre-trained models to specialized tasks. FrugalGPT [10] introduces strategies to reduce the computational and financial costs associated with deploying LLMs, focusing on optimizing model architectures and inference techniques to achieve cost-effective performance without significantly compromising accuracy. MacNet [47] utilizes directed acyclic graphs to organize LLM agents, streamlining their interactive reasoning via topological ordering.

# C  Algorithm of LessonL

The LessonL framework sketched in Algorithm 1 (Section 3) is elaborated in Algorithm 2 with details, together with subroutines Algorithms 3–5.

---

**Algorithm 2** The LessonL Framework

---

**Input:** code $x$; number of agents, $n$; number of rounds, $T$; number of selected lessons, $k$

1: $Z_{all} = \{\}$ ▷ initialize the lesson bank
2: **for** agent $j = 1, \ldots, n$ **do** ▷ initial round
3:     Get new code $y_j^{(0)} = \text{LLM}_j(x)$
4:     Get speedup $s_j^{(0)} = \text{Exec}(y_j^{(0)})$ and lesson $z_j^{(0)} = \text{LLM}_j(x, y_j^{(0)}, s_j^{(0)})$
5:     Initialize factor $f_j^{(0)} = 1$
6:     $Z_{all}.\text{add}((z_j^{(0)}, s_j^{(0)}, f_j^{(0)}))$ ▷ insert lesson with speedup and factor into the lesson bank
7: **end for**
8: **for** round $t = 1, \ldots, T$ **do**
9:     **if** $tn \leq k$, let $Z^{(t)} = Z_{all}$; **otherwise then** ▷ select lessons $Z^{(t)}$ used in the next round
10:         $Z_s, Z_{remain} = \text{SELECTHIGHSPEEDUP}(Z_{all}, \text{cnt} = \lceil k/2 \rceil)$ ▷ select half of the lessons
11:         $Z_q = \text{SELECTHIGHRELEVANCE}(Z_{remain}, \text{cnt} = \lfloor k/2 \rfloor)$ ▷ select another half
12:         $Z^{(t)} = Z_s \cup Z_q$ ▷ all selected lessons
13:     **end if**
14:     **for** agent $j = 1, \ldots, n$ **do** ▷ next round
15:         Get new code $y_j^{(t)} = \text{LLM}_j(x, Z^{(t)})$ ▷ use selected lessons $Z^{(t)}$ to prompt
16:         Get speedup $s_j^{(t)} = \text{Exec}(y_j^{(t)})$ and lesson $z_j^{(t)} = \text{LLM}_j(x, y_j^{(t)}, s_j^{(t)})$
17:         Initialize factor $f_j^{(t)} = 1$
18:         $Z_{all}.\text{add}((z_j^{(t)}, s_j^{(t)}, f_j^{(t)}))$ ▷ insert lesson with speedup and factor into the lesson bank
19:     **end for**
20:     $\text{ADJUSTFACTOR}(Z^{(t)}, \{s_j^{(t)}\}_j)$ ▷ adjust factors for selected lessons based on performance
21: **end for**
22: **return** best among $\{y_1^{(0)}, \ldots, y_n^{(0)}, \ldots, y_1^{(T)}, \ldots, y_n^{(T)}\}$

---

**Algorithm 3** SELECTHIGHSPEEDUP($Z_{all}$, cnt)

---

1: Sort lessons in $Z_{all} = \{(z, s, f)\}$ in descending order of $s \times f$   $\triangleright$ speedup $\times$ factor
2: Let $Z_s$ include top cnt lessons whose $s \times f \geq$ threshold
3: $Z_{remain} = Z_{all} \backslash Z_s$
4: **return** $Z_s, Z_{remain}$

---

**Algorithm 4** SELECTHIGHRELEVANCE($Z_{remain}$, cnt)

---

1: Sort lessons in $Z_{remain} = \{(z, s, f)\}$ in descending order of $\cos(z, x)$  $\triangleright$ $\cos(\text{lesson}, \text{code})$
2: Let $Z_q$ include top cnt lessons
3: **return** $Z_q$

---

**Algorithm 5** ADJUSTFACTOR($Z^{(t)}, \{s_j^{(t)}\}_j$)

---

1: **for** each lesson $(z, s, f)$ in $Z^{(t)}$ **do**
2:  Initialize correction $c \leftarrow 0$
3:  **for** agent $j = 1, \ldots, n$ **do**
4:    **if** $s < s_j^{(t)}$ **then**   $\triangleright$ original speedup $s <$ yielded speedup when applying the lesson
5:      Update correction $c \leftarrow c + (1 + \epsilon)$
6:    **else**
7:      Update correction $c \leftarrow c + (1 - \epsilon)$
8:    **end if**
9:  **end for**
10:  Update factor $f \leftarrow c/n$
11: **end for**

---

# D  Prompt Templates for Code Optimization

This section provides the prompt templates for code optimization. The initial prompt differs from subsequent prompts in that it starts with no lessons. The lesson solicitation prompt has several variants depending on the quality of the output code (speedup, slowdown, incorrect output, compilation error).

## D.1  Initial Prompt

> You are given a piece of code written in {language}. Your task is to rewrite it in the same language to improve its performance (i.e., execution time). {use_openmp} Do not change the input/output behaviors of the code. Include the generated code between "'{language} and "'.
>
> // Code:
> {source_code}

In our experiments, the variable language is "C++". The variable use_openmp is set to "You should use OpenMP to parallelize the code." for OpenMP prompts; otherwise it is empty.

## D.2  Lesson Solicitation

> The following are two functionally equivalent codes. They are compiled by using the same compiler and executed in the same environment. Code B runs {faster_or_slower} than Code A with a speedup {speedup}x. Explain why Code B is {faster_or_slower}. Be brief in the explanations. Use only one or two sentences.
>
> // Code A:

{source_code}

// Code B:
{target_code}

---

The following two codes are not functionally equivalent; that is, given the same input, they produce different outputs. Explain the reasons that make Code B nonequivalent to Code A. Be brief in the explanations. Use only one or two sentences.

// Code A:
{source_code}

// Code B:
{target_code}

---

The following are two codes. Code B attempts to improve the performance of Code A, but it has syntactic errors. Explain why Code B cannot be compiled. You may get hints from the compiler output provided after Code B. Be brief in the explanations. Use only one or two sentences.

// Code A:
{source_code}

// Code B:
{target_code}

Compiler output:
{compiler_output}

## D.3 Subsequent Prompt With Lessons

You are given a piece of code written in {language}. Your task is to rewrite it in the same language to improve its performance (i.e., execution time). {use_openmp} Do not change the input/output behaviors of the code. Some lessons regarding correctness and performance are provided to help you rewrite the code. Include the generated code between "'{language} and "'.

// Code:
{source_code}

While you rewrite the code, consider the following lessons. If Code A and Code B appear in the lessons, Code A refers to the given code and Code B refers to an attempted rewrite. Code B may not be optimal and it could be even worse than Code A.

{lesson[0]}

{lesson[1]}

{lesson[2]}

{lesson[3]}

Besides the above lessons, consider other optimization strategies that can more significantly improve the performance of the given code.

Each item in the lesson list can be one of the following:

- "Lesson {idx} slightly improves the code performance. {lesson_content} However, despite the code performance improvement, the speedup is only marginal."

- "Lesson {idx} significantly improves the code performance. {lesson_content}"

- "Lesson {idx} degrades the code performance. {lesson_content}"

- "Lesson {idx} compromises code equivalence. {lesson_content}"

- "Lesson {idx} produces non-compilable code. {lesson_content}"

# E  Prompt Templates for Code Generation

This section provides the prompt templates for code generation. The initial prompt differs from subsequent prompts in that it starts with no lessons. Following common practice, we include an example in the prompt, demonstrating the function signature, function description, and function completion.

## E.1  Initial Prompt

> You are given a function signature in {language} together with a docstring that explains what the function does. Your task is to implement the function according to the docstring. You should restate the function signature and docstring. Include the generated code between "'{language} and "'. For example, given the function signature and docstring
>
> {example_function_signature_and_docstring}
>
> You should respond with
>
> {example_generation}
>
> ### Here is the function to implement:
> {function_signature_and_docstring}

In our experiments, the variable language is "Python".

The variable example_function_signature_and_docstring is

```
def sum(a: float, b: float) -> float:
    """ Return the sum of two floats a and b """
```

The variable example_generation is

```
'''Python
def sum(a: float, b: float) -> float:
    """ Return the sum of two floats a and b """
    return a + b
'''
```

### E.2 Lesson Solicitation

> The following completed code is incorrect; i.e., it does not exactly reflect the description in the docstring. The code passes only {num_pass_cases} test cases out of {num_total_cases}, leaving {num_fail_cases} failed. Explain why the code is incorrect (that is, why it fails some test cases). Be brief in the explanations. Use only one or two sentences.
>
> ### Completed code:
> {completed_code}

### E.3 Subsequent Prompt With Lessons

> You are given a function signature in {language} together with a docstring that explains what the function does. Your task is to implement the function according to the docstring. You should restate the function signature and docstring. Some lessons are provided to help you implement the function. Include the generated code between "`{language}` and "`. For example, given the function signature and docstring
>
> {example_function_signature_and_docstring}
>
> You should respond with
>
> {example_generation}
>
> ### Here is the function to implement:
> {function_signature_and_docstring}
>
> While you implement the function, consider the following lessons.
>
> {lesson[0]}
>
> {lesson[1]}
>
> {lesson[2]}
>
> {lesson[3]}

Each item in the lesson list is:

• "Lesson {idx} reasons why the code does not pass all test cases. {lesson_content}"

## F  ParEval Modification for Code Optimization

We run experiments on a code optimization benchmark adapted from ParEval [39], which is an existing benchmark for LLM code generation. At a high level, our code optimization benchmark tasks an LLM with optimizing a piece of source code. The LLM's response is evaluated based on its correctness as well as the speedup it can achieve over the source code. In this section, we discuss our adaptation and improvement of ParEval.

### F.1  Adaptation from ParEval

In the original ParEval benchmark, the LLM is tasked with completing a function definition. To test correctness, the LLM generated code is run on custom test cases and it is compared to a provided baseline.

In our code optimization benchmark, we use the baseline as the source code to be optimized. Some changes are made to the source code for particular problems, such as adding definitions of custom types, so that the source code can run on its own.

### F.2  Benchmarking

ParEval performs benchmarking by taking the average runtime of the baseline code and the LLM generated code over a fixed number of runs. This approach can produce noisy results if the execution time of the baseline code or LLM generated code is very short. To address this issue, we use Nanobench, a lightweight benchmarking library [3]. Nanobench varies the number of benchmarking runs based on the runtime of the code as well as its variance, thereby producing stable results on codes with shorter runtimes. Overall, Nanobench produces consistent and stable results, which ensures that LLMs are accurately evaluated on the benchmark.

### F.3  Input Generation

We ensure that the randomness used when generating inputs for the baseline code and the LLM generated code are the same. Fixing randomness among the two is necessary when benchmarking as the runtime of certain problems can vary greatly depending on the specific inputs provided. For example, consider the task of finding a value in a list. If the value is the first element in the list, then the runtime will be much faster compared to when the value is not in the list. We ensure all inputs are generated from a seeded random generator, and ensure that the seed is the same when running the baseline and when running the LLM generated code.

We also improve the random input generation for some problems to ensure that all cases are considered and thoroughly tested. First, we look at improving the input generation for graph problems, which includes finding the number of connected components in a graph, finding the size of the largest connected component in the graph, and finding the shortest path length in a graph. Previously, ParEval generated random graphs using the $G(n, p)$ Erdos-Renyi model. Each edge in the $n$-vertex graph would be independently added with probability $p$. The issue with this type of generation using a fixed probability $p$ for large graphs is that the resulting graph is almost surely connected, which means the number of connected components is $1$, the size of the largest connected component is $n$, and there always exists a path between any two vertices. Therefore, some incorrect codes can pass the provided test cases as the test cases are not varied enough. Therefore, to mitigate this issue, we vary the distribution of generated graphs by implementing a simple random graph algorithm, which first randomly selects the number of edges in the graph, and then for each edge randomly selects the two vertices incident to the edge. We limit the number of edges in the graphs to generate graphs with a wide range of connected components for the purposes of these problems. Alternative random graph generation algorithms can be used, but we find this simple algorithm works for the specific graph problems in the benchmark.

Next, we look at improving the input generation for a search problem which tasks the user with finding a target value in a list. The current code from ParEval generates random inputs from a narrow range of values, which means that when benchmarking on large input sizes, all values in the range will be generated, which has the effect that any target value will almost surely be in the list. This allows incorrect codes, such as a function that always returns true, to potentially pass the provided test cases on large input sizes.

### F.4  Other Fixes

We also fix several bugs in the existing ParEval benchmark when adapting it to the code optimization setting. Several problems encounter overflow issues when scaling up the input size, as the generated input values are quite large. We fix this behavior by changing the range of random values generated for these specific problems. In addition, we fix bugs in the baseline codes of several problems that produce incorrect results, such as dense and sparse matrix LU decomposition, finding the convex hull and computing its perimeter, and computing the maximum contiguous subarray sum.

## G  Additional Information of Evaluation Setup

### G.1  Baseline Implementation for Code Optimization

CoT: We prompt the agent by adding "Let's think step by step" at the end to solicit reasoning.

Reflexion: Reflexion was originally used for code generation, wherein each iteration, synthetic test cases are constructed and verbal feedback of the generated code is produced based on the code's correctness on these test cases. The iterations will terminate when the generated code is correct. For code optimization, we replace the testing of correctness with the measurement of speedup. We have Reflexion run ten iterations and choose the code with the best speedup over these iterations as the final solution.

MapCoder: MapCoder was originally used for code generation. Given a coding task, MapCoder first retrieves similar tasks as examples, then proposes a few plans to solve the current task, and executes the plans one-by-one in the decreasing order of its confidence. For every plan, a few iterations of coding–debugging are performed. MapCoder will terminate when a plan leads to correct code. For code optimization, we replace the testing of correctness in the debugging step with the measurement of speedup. We have MapCoder produce three plans and run five iterations for each plan. We choose the code with the best speedup over all plans and all iterations as the final solution.

MoA: We follow the default configuration, using three layers and having GPT-4o as the aggregator.

## G.2 Baseline Implementation for Code Generation

Different from ParEval, the benchmarks for code generation offer a fixed set of test cases. HumanEval/MBPP offer very few test cases (which is a motivation of developing HumanEval+/MBPP+). Hence, in some of the baselines, there is a choice between using the given set of test cases and using syntheticly generated test cases in each iteration. The drawback of using the given set is that the generated code may overfit the test cases when the set is small, whereas the drawback of using a synthetic set, which is generated by an LLM, is that the test cases may be incorrect. We consider overfit doing more harm and use syntheticly generated test cases instead. Following Reflexion [49], each time, six test cases are generated by Qwen14B.

For all four benchmarks, we use EvalPlus [36] to evaluate the generation.

LessonL: As opposed to the code optimization case, for code generation, LessonL terminates the iterations whenever correct code, as measured by the synthetic test cases, is produced. The pass@1 score is reported by using the given test cases.

CoT: Same as the configuration in the code optimization case.

Reflexion: We use a maximum of ten iterations.

MapCoder: We use three plans and a maximum of five iterations each.

MoA: Same as the configuration in the code optimization case.

## G.3 Hyperparameters, Hardware, and Timing

**Hyperparameters.** The hyperparameters used by LessonL are summarized in Table 6, unless otherwise stated (e.g., in ablation studies). We barely tune them. Additionally, LLM parameters for all models are also included in Table 6. Both code optimization and code generate tasks share the same set of hyperparameters, except that code generation does not use SELECTHIGHSPEEDUP.

Table 6: Hyperparameters.

| | |
|---|---|
| Number of agents, $n$ | 3 |
| Number of rounds, $T$ | 4 |
| Number of selected lessons, $k$ | 4 |
| Threshold in SELECTHIGHSPEEDUP | 1.1 |
| $\epsilon$ in ADJUSTFACTOR | 0.1 |
| LLM temperature | 0.2 |
| LLM frequency penalty | 0.5 |

**Hardware.** For code optimization benchmarks (ParEval and PolyBench), the LLMs are hosted on an Amazon EC2 g6e.12xlarge instance through vLLM [31]. This instance contains 4x Nvidia L40S GPUs (192 GB). Additionally, we use c5.4xlarge instances, each with 16x Intel Xeon Platinum

vCPUs (3.4 GHz), to ensure resource isolation from the g6e.12xlarge instance when evaluating code correctness and speedup. All baselines and methods use one single c5.4xlarge exclusively for performance evalution to ensure consistent hardware resources. For code generation (HumanEval, HumanEval+, MBPP, MBPP+), we conduct experiments on a server with eight A6000 GPUs. The LLMs are hosted on these resources.

**Timing.** A full run of LessonL on ParEval takes a few hours. The runtime for other methods and/or other benchmarks falls in a similar scale.

# H   Supplementary Information for Figure 3

Figure 3 in the main text shows the performance variation as the number of rounds increases for methods that do allow iterations (MoA, MapCoder, and Reflexion). In MoA, they are called "layers." While prior work suggests that these methods' performance may improve with more iterations, we find that it does not necessarily happen for code optimization. In what follows, for each method (including LessonL), we show an example of the output code at each round and discuss the speedup variation. For readability, the codes were edited by removing nonessential comments (such as problem description) and white spaces (including indentation).

## H.1   LessonL

Below are the outputs of LessonL. The speedup generally improves over rounds. The code is to perform convolution on an image. In round 0, the code is improved by removing the redundant zero-additions. The code in round 1 is identical to that in round 0. In round 2, the code handles the interior region and the boundary separately, yielding further improvement. Round 3 achieves a significant further improvement by removing the temporary storage and avoiding extra data copies. The code in round 4 is identical to that in round 3.

---

**Original code**

```
#pragma once
#include <vector>

const int edgeKernel[3][3] = {{-1, -1, -1}, {-1, 8, -1}, {-1, -1, -1}};

/* Convolve the edge kernel with a grayscale image. Each pixel will be
replaced with
   the dot product of itself and its neighbors with the edge kernel.
   Use a value of 0 for pixels outside the image's boundaries and clip
   outputs between 0 and 255.
   imageIn and imageOut are NxN grayscale images stored in row-major.
   Store the output of the computation in imageOut.
   Example:
   input: [[112, 118, 141, 152],
   [93, 101, 119, 203],
   [45, 17, 16, 232],
   [82, 31, 49, 101]]
   output: [[255, 255, 255, 255],
   [255, 147, 0, 255],
   [36, 0, 0, 255],
   [255, 39, 0, 255]]
*/
void convolveKernel(std::vector<int> const& imageIn, std::vector<int>
&imageOut, size_t N) {
  for (size_t i = 0; i < N; i++) {
    for (size_t j = 0; j < N; j++){
      int sum = 0;
      for (int k = -1; k < 2; k++) {
        for (int l = -1; l < 2; l++){
```

---

```
            int x = i + k;
            int y = j + l;
            if ((x < 0) || (x >= N) || (y < 0) || (y >= N)) {
              sum += 0;
            } else {
              sum += imageIn[x * N + y] * edgeKernel[k + 1][l + 1];
            }
          }
        }
        if (sum < 0) {
          imageOut[i * N + j] = 0;
        } else if (sum > 255) {
          imageOut[i * N + j] = 255;
        } else {
          imageOut[i * N + j] = sum;
        }
      }
    }
}
```

**LessonL, round 0, speedup 1.71x**

```
#pragma once
#include <vector>
#include <algorithm>

const int edgeKernel[3][3] = {{-1, -1, -1}, {-1, 8, -1}, {-1, -1, -1}};

void convolveKernel(std::vector<int> const& imageIn, std::vector<int>
&imageOut, size_t N) {
  for (size_t i = 0; i < N; i++) {
    for (size_t j = 0; j < N; j++) {
      int sum = 0;
      for (int k = -1; k < 2; k++) {
        for (int l = -1; l < 2; l++) {
          int x = i + k;
          int y = j + l;
          if (x >= 0 && x < N && y >= 0 && y < N) {
            sum += imageIn[x * N + y] * edgeKernel[k + 1][l + 1];
          }
        }
      }
      imageOut[i * N + j] = std::clamp(sum, 0, 255);
    }
  }
}
```

**LessonL, round 1, speedup 1.71x**

```
#pragma once
#include <vector>
#include <algorithm> // For std::clamp

const int edgeKernel[3][3] = {{-1, -1, -1}, {-1, 8, -1}, {-1, -1, -1}};

void convolveKernel(std::vector<int> const& imageIn, std::vector<int>
&imageOut, size_t N) {
  for (size_t i = 0; i < N; i++) {
    for (size_t j = 0; j < N; j++) {
```

```
      int sum = 0;
      for (int k = -1; k < 2; k++) {
        for (int l = -1; l < 2; l++) {
          int x = i + k;
          int y = j + l;
          if (x >= 0 && x < N && y >= 0 && y < N) {
            sum += imageIn[x * N + y] * edgeKernel[k + 1][l + 1];
          }
        }
      }
      imageOut[i * N + j] = std::clamp(sum, 0, 255);
    }
  }
}
```

**LessonL, round 2, speedup 2.09x**

```
#pragma once
#include <vector>
#include <algorithm> // For std::clamp

const int edgeKernel[3][3] = {{-1, -1, -1}, {-1, 8, -1}, {-1, -1, -1}};

void convolveKernel(std::vector<int> const& imageIn, std::vector<int>
&imageOut, size_t N) {
  // Temporary vector to store intermediate results
  std::vector<int> temp(N * N, 0);

  // Process the interior pixels
  for (size_t i = 1; i < N - 1; i++) {
    for (size_t j = 1; j < N - 1; j++) {
      int sum = 0;
      for (int k = -1; k < 2; k++) {
        for (int l = -1; l < 2; l++) {
          sum += imageIn[(i + k) * N + (j + l)] * edgeKernel[k + 1][l
          + 1];
        }
      }
      temp[i * N + j] = sum;
    }
  }

  // Process the boundary pixels
  for (size_t i = 0; i < N; i++) {
    for (size_t j = 0; j < N; j++) {
      if (i == 0 || i == N - 1 || j == 0 || j == N - 1) {
        int sum = 0;
        for (int k = -1; k < 2; k++) {
          for (int l = -1; l < 2; l++) {
            int x = i + k;
            int y = j + l;
            if (x >= 0 && x < N && y >= 0 && y < N) {
              sum += imageIn[x * N + y] * edgeKernel[k + 1][l + 1];
            }
          }
        }
        temp[i * N + j] = sum;
      }
    }
  }
```

```
  // Apply clipping to the results
  for (size_t i = 0; i < N * N; i++) {
    imageOut[i] = std::clamp(temp[i], 0, 255);
  }
}
```

**LessonL, round 3, speedup 9.45x**

```
#pragma once
#include <vector>
#include <algorithm> // For std::clamp

const int edgeKernel[3][3] = {{-1, -1, -1}, {-1, 8, -1}, {-1, -1, -1}};

void convolveKernel(std::vector<int> const& imageIn, std::vector<int>
&imageOut, size_t N) {
  // Process interior pixels
  for (size_t i = 1; i < N - 1; i++) {
    for (size_t j = 1; j < N - 1; j++) {
      int sum = 0;
      for (int k = -1; k < 2; k++) {
        for (int l = -1; l < 2; l++) {
          sum += imageIn[(i + k) * N + (j + l)] * edgeKernel[k + 1][l
          + 1];
        }
      }
      imageOut[i * N + j] = std::clamp(sum, 0, 255);
    }
  }

  // Process boundary pixels
  for (size_t i = 0; i < N; i++) {
    for (size_t j = 0; j < N; j++) {
      if (i == 0 || i == N - 1 || j == 0 || j == N - 1) {
        int sum = 0;
        for (int k = -1; k < 2; k++) {
          for (int l = -1; l < 2; l++) {
            int x = i + k;
            int y = j + l;
            if ((x < 0) || (x >= N) || (y < 0) || (y >= N)) {
              sum += 0;
            } else {
              sum += imageIn[x * N + y] * edgeKernel[k + 1][l + 1];
            }
          }
        }
        imageOut[i * N + j] = std::clamp(sum, 0, 255);
      }
    }
  }
}
```

**LessonL, round 4, speedup 9.50x**

```
#pragma once
#include <vector>
#include <algorithm> // For std::clamp

const int edgeKernel[3][3] = {{-1, -1, -1}, {-1, 8, -1}, {-1, -1, -1}};
```

```
void convolveKernel(std::vector<int> const& imageIn, std::vector<int>
&imageOut, size_t N) {
  // Process interior pixels
  for (size_t i = 1; i < N - 1; i++) {
    for (size_t j = 1; j < N - 1; j++) {
      int sum = 0;
      for (int k = -1; k < 2; k++) {
        for (int l = -1; l < 2; l++) {
          sum += imageIn[(i + k) * N + (j + l)] * edgeKernel[k + 1][l
          + 1];
        }
      }
      imageOut[i * N + j] = std::clamp(sum, 0, 255);
    }
  }

  // Process boundary pixels
  for (size_t i = 0; i < N; i++) {
    for (size_t j = 0; j < N; j++) {
      if (i == 0 || i == N - 1 || j == 0 || j == N - 1) {
        int sum = 0;
        for (int k = -1; k < 2; k++) {
          for (int l = -1; l < 2; l++) {
            int x = i + k;
            int y = j + l;
            if ((x < 0) || (x >= N) || (y < 0) || (y >= N)) {
              sum += 0;
            } else {
              sum += imageIn[x * N + y] * edgeKernel[k + 1][l + 1];
            }
          }
        }
        imageOut[i * N + j] = std::clamp(sum, 0, 255);
      }
    }
  }
}
```

## H.2  MoA

Below are the outputs of MoA. The speedup deteriorates over rounds/layers. The code finds the $k$th smallest element of an array. The code in round 0 achieves the highest speedup by implementing the divide-and-conquer algorithm. In round 1, the speedup reduces because of the type cast from time_t to unsigned int. In round 2, the speedup further reduces because the implementation is changed from divide-and-conquer to using std::nth_element. The codes in round 3, 4, and 5 are identical to that in round 2.

Original code

```
#pragma once
#include <algorithm>
#include <vector>

/* Find the k-th smallest element of the vector x.
   Example:

   input: x=[1, 7, 6, 0, 2, 2, 10, 6], k=4
   output: 6
```

```
*/
int findKthSmallest(std::vector<int> const& x, int k) {
  std::vector<int> x_copy = x;
  std::sort(x_copy.begin(), x_copy.end());
  return x_copy[k-1];
}
```

**MoA, round 0, speedup 12.36x**

```
#pragma once
#include <vector>
#include <cstdlib> // For std::rand() and std::srand()

int partition(std::vector<int>& x, int left, int right, int
pivotIndex) {
  int pivotValue = x[pivotIndex];
  std::swap(x[pivotIndex], x[right]); // Move pivot to end
  int storeIndex = left;
  for (int i = left; i < right; ++i) {
    if (x[i] < pivotValue) {
      std::swap(x[storeIndex], x[i]);
      ++storeIndex;
    }
  }
  std::swap(x[right], x[storeIndex]); // Move pivot to its final place
  return storeIndex;
}

int quickselect(std::vector<int>& x, int left, int right, int k) {
  if (left == right) { // If the list contains only one element,
    return x[left];  // return that element
  }

  // Select a random pivotIndex
  int pivotIndex = left + std::rand() % (right - left + 1);

  // Find the pivot position in a sorted list
  pivotIndex = partition(x, left, right, pivotIndex);

  // The pivot is in its final sorted position
  if (k == pivotIndex) {
    return x[k];
  } else if (k < pivotIndex) {
    return quickselect(x, left, pivotIndex - 1, k);
  } else {
    return quickselect(x, pivotIndex + 1, right, k);
  }
}

int findKthSmallest(std::vector<int> const& x, int k) {
  std::vector<int> x_copy = x;
  std::srand(time(0)); // Seed the random number generator
  return quickselect(x_copy, 0, x_copy.size() - 1, k - 1);
}
```

**MoA, round 1, speedup 8.10x**

```
#pragma once
#include <vector>
```

```cpp
#include <cstdlib> // For std::rand() and std::srand()
#include <ctime>   // For std::time()

int partition(std::vector<int>& x, int left, int right, int
pivotIndex) {
  int pivotValue = x[pivotIndex];
  std::swap(x[pivotIndex], x[right]); // Move pivot to end
  int storeIndex = left;
  for (int i = left; i < right; ++i) {
    if (x[i] < pivotValue) {
      std::swap(x[storeIndex], x[i]);
      ++storeIndex;
    }
  }
  std::swap(x[right], x[storeIndex]); // Move pivot to its final place
  return storeIndex;
}

int quickselect(std::vector<int>& x, int left, int right, int k) {
  if (left == right) { // If the list contains only one element,
    return x[left];  // return that element
  }

  // Select a random pivotIndex
  int pivotIndex = left + std::rand() % (right - left + 1);

  // Find the pivot position in a sorted list
  pivotIndex = partition(x, left, right, pivotIndex);

  // The pivot is in its final sorted position
  if (k == pivotIndex) {
    return x[k];
  } else if (k < pivotIndex) {
    return quickselect(x, left, pivotIndex - 1, k);
  } else {
    return quickselect(x, pivotIndex + 1, right, k);
  }
}

int findKthSmallest(std::vector<int> const& x, int k) {
  std::vector<int> x_copy = x;
  std::srand(static_cast<unsigned int>(std::time(0))); // Seed the
  random number generator
  return quickselect(x_copy, 0, x_copy.size() - 1, k - 1);
}
```

**MoA, round 2, speedup 6.71x**

```cpp
#pragma once
#include <vector>
#include <algorithm>

int findKthSmallest(std::vector<int> const& x, int k) {
  std::vector<int> x_copy = x;
  std::nth_element(x_copy.begin(), x_copy.begin() + k - 1,
  x_copy.end());
  return x_copy[k - 1];
}
```

**MoA, round 3, speedup 6.72x**

```
#pragma once
#include <vector>
#include <algorithm>

int findKthSmallest(std::vector<int> const& x, int k) {
  std::vector<int> x_copy = x;
  std::nth_element(x_copy.begin(), x_copy.begin() + k - 1,
  x_copy.end());
  return x_copy[k - 1];
}
```

**MoA, round 4, speedup 6.74x**

```
#pragma once
#include <algorithm>
#include <vector>

int findKthSmallest(std::vector<int> const& x, int k) {
  std::vector<int> x_copy = x;
  std::nth_element(x_copy.begin(), x_copy.begin() + k - 1,
  x_copy.end());
  return x_copy[k - 1];
}
```

**MoA, round 5, speedup 6.72x**

```
#pragma once
#include <algorithm>
#include <vector>

int findKthSmallest(std::vector<int> const& x, int k) {
  std::vector<int> x_copy = x;
  std::nth_element(x_copy.begin(), x_copy.begin() + k - 1,
  x_copy.end());
  return x_copy[k - 1];
}
```

## H.3  MapCoder

Below are the outputs of MapCoder. The speedup does not follow a monotonic trend over rounds. MapCoder works by using one agent to generate a few plans and evaluate each plan with a confidence rating. The plans are executed in the decreasing order of confidence, but it does not necessarily mean that the earlier executed plans will lead to better performing codes. In this example, the original code is to compute the highest node degree in an undirected graph. This amounts to computing the sum of each row of the adjacency and comparing different rows. All the plans perform some sort of loop unrolling, the idea of which is gaining less attraction nowadays because loop unrolling should be delegated to the compiler for better instruction parallelism and fewer comparisons. As a result, none of the output codes runs faster than the original code.

**Oiginal code**

```
#pragma once
#include <algorithm>
#include <vector>
#include <queue>
```

```
#include <stack>
#include <deque>
#include <limits>

/* Compute the highest node degree in the undirected graph. The graph
is defined in the adjacency matrix A.
   A is an NxN adjacency matrix stored in row-major. A is undirected.
   Example:
   input: [[0, 0, 0, 1], [0, 0, 0, 1], [0, 0, 0, 1], [1, 1, 1, 0]]
   output: 3
*/
int maxDegree(std::vector<int> const& A, size_t N) {
  int highestDegree = 0;
  for (int i = 0; i < N; i += 1) {
    int degree = 0;
    for (int j = 0; j < N; j += 1) {
      degree += A[i * N + j];
    }
    highestDegree = std::max(highestDegree, degree);
  }
  return highestDegree;
}
```

**MapCoder, round 0, speedup 1.00x**

```
#pragma once
#include <algorithm>
#include <vector>
#include <numeric> // For std::accumulate

int maxDegree(const std::vector<int>& A, size_t N) {
  int highestDegree = 0;
  for (size_t i = 0; i < N; i += 2) { // Loop Unrolling: Process two
  rows at a time
    int degree1 = std::accumulate(A.begin() + i * N, A.begin() + (i +
    1) * N, 0);
    int degree2 = (i + 1 < N) ? std::accumulate(A.begin() + (i + 1) *
    N, A.begin() + (i + 2) * N, 0) : 0;
    highestDegree = std::max(highestDegree, std::max(degree1,
    degree2));
  }
  return highestDegree;
}
```

**MapCoder, round 1, speedup 0.96x**

```
#pragma once
#include <algorithm>
#include <vector>
#include <queue>
#include <stack>
#include <deque>
#include <limits>

inline int maxDegree(const std::vector<int>& A, size_t N) {
  int highestDegree = 0;
  for (size_t i = 0; i < N; i += 4) { // Loop unrolling by a factor of
  4
    int degree0 = 0, degree1 = 0, degree2 = 0, degree3 = 0;
```

```
    if (i + 0 < N) {
      for (size_t j = 0; j < N; j += 4) { // Inner loop unrolling by a
      factor of 4
        degree0 += A[(i + 0) * N + j + 0];
        degree0 += A[(i + 0) * N + j + 1];
        degree0 += A[(i + 0) * N + j + 2];
        degree0 += A[(i + 0) * N + j + 3];
        if (i + 1 < N) {
          degree1 += A[(i + 1) * N + j + 0];
          degree1 += A[(i + 1) * N + j + 1];
          degree1 += A[(i + 1) * N + j + 2];
          degree1 += A[(i + 1) * N + j + 3];
        }
        if (i + 2 < N) {
          degree2 += A[(i + 2) * N + j + 0];
          degree2 += A[(i + 2) * N + j + 1];
          degree2 += A[(i + 2) * N + j + 2];
          degree2 += A[(i + 2) * N + j + 3];
        }
        if (i + 3 < N) {
          degree3 += A[(i + 3) * N + j + 0];
          degree3 += A[(i + 3) * N + j + 1];
          degree3 += A[(i + 3) * N + j + 2];
          degree3 += A[(i + 3) * N + j + 3];
        }
      }
    }
    highestDegree = std::max({highestDegree, degree0, degree1,
    degree2, degree3});
  }
  return highestDegree;
}
```

**MapCoder, round 2, speedup 0.65x**

```
#pragma once
#include <algorithm>
#include <vector>
#include <queue>
#include <stack>
#include <deque>
#include <limits>

int maxDegree(const std::vector<int>& A, size_t N) {
  int highestDegree = 0;
  for (size_t i = 0; i < N; i += 1) {
    int degree = 0;
    // Unrolling the inner loop
    for (size_t j = 0; j < N; j += 4) {
      if (j + 0 < N) degree += A[i * N + j + 0];
      if (j + 1 < N) degree += A[i * N + j + 1];
      if (j + 2 < N) degree += A[i * N + j + 2];
      if (j + 3 < N) degree += A[i * N + j + 3];
    }
    highestDegree = std::max(highestDegree, degree);
  }
  return highestDegree;
}
```

**MapCoder, round 3, speedup 0.74x**

```
#pragma once
#include <algorithm>
#include <vector>
#include <queue>
#include <stack>
#include <deque>
#include <limits>

int maxDegree(std::vector<int> const& A, size_t N) {
  int highestDegree = 0;
  for (int i = 0; i < N; i += 1) {
    int degree = 0;
    for (int j = 0; j < N; j += 4) { // Unroll the loop by a factor of
    4
      degree += A[i * N + j];
      if (j + 1 < N) degree += A[i * N + j + 1];
      if (j + 2 < N) degree += A[i * N + j + 2];
      if (j + 3 < N) degree += A[i * N + j + 3];
    }
    highestDegree = std::max(highestDegree, degree);
  }
  return highestDegree;
}
```

**MapCoder, round 4, speedup 1.00x**

```
#pragma once
#include <algorithm>
#include <vector>
#include <queue>
#include <stack>
#include <deque>
#include <limits>

int maxDegree(std::vector<int> const& A, size_t N) {
  int highestDegree = 0;
  for (int i = 0; i < N; i += 1) {
    int degree = 0;
    int j = 0;
    // Unroll the inner loop by a factor of 4
    for (; j <= N - 4; j += 4) {
      degree += A[i * N + j];
      degree += A[i * N + j + 1];
      degree += A[i * N + j + 2];
      degree += A[i * N + j + 3];
    }
    // Handle the remaining elements
    for (; j < N; j += 1) {
      degree += A[i * N + j];
    }
    highestDegree = std::max(highestDegree, degree);
  }
  return highestDegree;
}
```

## H.4 Reflexion

Below are the outputs of Reflexion. The speedup generally improves over rounds. The code is to perform cell XOR. The code in round 0 fails to compile. The code in round 1 is identical to the original code. The code in round 2 is marginally slower than the original code. This is likely because

of the timing noise; in fact, this code precomputes the common expression 'i * N + j', which can slightly improves the runtime. In round 3, the code performance drastically improves, because half of the if-tests are eliminated. The code in round 4 is identical to that of round 3.

**Original code**

```cpp
#pragma once
#include <vector>

/* Set every cell's value to 1 if it has exactly one neighbor that's a
1. Otherwise set it to 0.
   Note that we only consider neighbors and not input_{i,j} when
   computing output_{i,j}.
   input and output are NxN grids of ints in row-major.
   Example:
   input: [[0, 1, 1, 0],
   [1, 0, 0, 0],
   [0, 0, 0, 0],
   [0, 1, 0, 0]
   output: [[0, 0, 1, 1],
   [1, 0, 0, 1],
   [0, 0, 1, 0],
   [1, 0, 1, 0]]
*/
void cellsXOR(std::vector<int> const& input, std::vector<int> &output,
size_t N) {
  for (size_t i = 0; i < N; ++i) {
    for (size_t j = 0; j < N; ++j) {
      int count = 0;
      if (i > 0 && input[(i-1)*N + j] == 1) ++count;
      if (i < N-1 && input[(i+1)*N + j] == 1) ++count;
      if (j > 0 && input[i*N + j-1] == 1) ++count;
      if (j < N-1 && input[i*N + j+1] == 1) ++count;
      output[i*N + j] = (count == 1) ? 1 : 0;
    }
  }
}
```

**Reflexion, round 0, compilation failed**

```cpp
#pragma once
#include <vector>

// Lookup table for the count of neighbors
const int neighborCountLookup[9] = {
  0, 0, 0, // 0 neighbors
  1, 0, 0, // 1 neighbor
  0, 0, 0, // 2 neighbors
  0, 0, 0, // 3 neighbors
  0, 0, 0, // 4 neighbors
  0, 0, 0, // 5 neighbors
  0, 0, 0, // 6 neighbors
  0, 0, 0, // 7 neighbors
  0, 0, 0  // 8 neighbors
};

void cellsXOR(std::vector<int> const& input, std::vector<int> &output,
size_t N) {
  for (size_t i = 0; i < N; ++i) {
    for (size_t j = 0; j < N; ++j) {
```

```
        int count = 0;
        if (i > 0) {
          if (j > 0) count += input[(i-1)*N + j-1];
          count += input[(i-1)*N + j];
          if (j < N-1) count += input[(i-1)*N + j+1];
        }
        if (j > 0) count += input[i*N + j-1];
        if (j < N-1) count += input[i*N + j+1];
        if (i < N-1) {
          if (j > 0) count += input[(i+1)*N + j-1];
          count += input[(i+1)*N + j];
          if (j < N-1) count += input[(i+1)*N + j+1];
        }
        output[i*N + j] = neighborCountLookup[count];
      }
    }
}
```

```
#pragma once
#include <vector>

void cellsXOR(std::vector<int> const& input, std::vector<int> &output,
size_t N) {
  for (size_t i = 0; i < N; ++i) {
    for (size_t j = 0; j < N; ++j) {
      int count = 0;
      if (i > 0 && input[(i-1)*N + j] == 1) ++count;
      if (i < N-1 && input[(i+1)*N + j] == 1) ++count;
      if (j > 0 && input[i*N + j-1] == 1) ++count;
      if (j < N-1 && input[i*N + j+1] == 1) ++count;
      output[i*N + j] = (count == 1) ? 1 : 0;
    }
  }
}
```

```
#pragma once
#include <vector>

void cellsXOR(std::vector<int> const& input, std::vector<int> &output,
size_t N) {
  for (size_t i = 0; i < N; ++i) {
    for (size_t j = 0; j < N; ++j) {
      int count = 0;
      size_t idx = i * N + j;
      if (i > 0) {
        if (input[idx - N] == 1) ++count;
      }
      if (i < N - 1) {
        if (input[idx + N] == 1) ++count;
      }
      if (j > 0) {
        if (input[idx - 1] == 1) ++count;
      }
      if (j < N - 1) {
        if (input[idx + 1] == 1) ++count;
```

```
    }
      output[idx] = (count == 1) ? 1 : 0;
    }
  }
}
```

```cpp
#pragma once
#include <vector>

void cellsXOR(std::vector<int> const& input, std::vector<int> &output,
size_t N) {
  for (size_t i = 0; i < N; ++i) {
    for (size_t j = 0; j < N; ++j) {
      int count = 0;
      size_t idx = i * N + j;
      if (i > 0) {
        count += input[idx - N];
      }
      if (i < N - 1) {
        count += input[idx + N];
      }
      if (j > 0) {
        count += input[idx - 1];
      }
      if (j < N - 1) {
        count += input[idx + 1];
      }
      output[idx] = (count == 1) ? 1 : 0;
    }
  }
}
```

```cpp
#pragma once
#include <vector>

void cellsXOR(std::vector<int> const& input, std::vector<int> &output,
size_t N) {
  for (size_t i = 0; i < N; ++i) {
    for (size_t j = 0; j < N; ++j) {
      int count = 0;
      size_t idx = i * N + j;
      if (i > 0) {
        count += input[idx - N];
      }
      if (i < N - 1) {
        count += input[idx + N];
      }
      if (j > 0) {
        count += input[idx - 1];
      }
      if (j < N - 1) {
        count += input[idx + 1];
      }
      output[idx] = (count == 1) ? 1 : 0;
    }
```

```
    }
}
```

# I  Supplementary Information for Figure 4

Figure 4 in the main text shows the monetary cost and FLOPS for various models and methods. For this, several pieces of information need to be estimated: the unit cost, FLOPS per token, and the number of tokens.

The pricing information for the open-source models is based on `https://www.together.ai/pricing` and that for the GPT models is from `https://platform.openai.com/docs/pricing`. Specifically, we estimate the unit cost of Deepseek7B, Qwen7B, Qwen14B, and DeepseekR1-14B to be \$0.20, \$0.20, \$0.30, and \$0.18, respectively, per 1M tokens. Meanwhile, GPT-4o mini costs \$0.15/\$0.60 per 1M input/output tokens, while GPT-4o costs \$2.50/\$10.00 and OpenAI o3 costs \$2.00/\$8.00. The pricing information was retrieved in April 2025 (except for DeepseekR1-14B and OpenAI o3) and it may be time-sensitive. The pricing for DeepseekR1-14B and OpenAI o3 was retrieved in October 2025 and it appears that the pricing is compatible with that of other models. The number of tokens is estimated based on the CodeLlama-7b-Instruct-hf tokenizer from Hugging-face [55]. For multi-agent methods, we assume each agent consumes/generates the same amount of tokens. Table 7 summarizes the information for one experiment on the ParEval benchmark.

Table 7: (Estimated) Costs for running a ParEval experiment. For multi-agent methods, the listed number of tokens is the sum over all agents. For MoA, the first row shows the number of tokens for the open-source agents, while the second row for the GPT-4o aggregator (separately priced). For CoT, Reflexion, and MapCoder, all agents are Qwen14B.

|  | Speedup | #Input tokens | #Output tokens | #Total tokens | Cost (\$) |
|---|---|---|---|---|---|
| Qwen14B(1) | 1.60 | 23421 | 33550 | 56971 | 0.017 |
| Qwen14B(20) | 1.70 | 468420 | 671001 | 1139421 | 0.342 |
| GPT-4o mini | 1.57 | 20815 | 26399 | 47214 | 0.019 |
| GPT-4o | 1.72 | 20815 | 27800 | 48615 | 0.330 |
| DeepseekR1-14B | 1.59 | 24900 | 107733 | 132633 | 0.024 |
| OpenAI o3 | 2.21 | 20875 | 98719 | 119594 | 0.832 |
| CoT(1) | 1.57 | 23681 | 37119 | 60800 | 0.018 |
| CoT(20) | 1.67 | 477220 | 742380 | 1219600 | 0.366 |
| Reflexion | 1.88 | 1126390 | 389520 | 1515910 | 0.455 |
| MapCoder | 1.85 | 1210143 | 713383 | 1923526 | 0.577 |
| MoA | 1.76 | 344316 | 172973 | 517289 | 0.612 |
|  |  | 79005 | 29441 | 108446 |  |
| LessonL | 2.16 | 1009359 | 387994 | 1397353 | 0.326 |

For FLOPS per token, we take the approach of [30] and estimate it to be $24 n_{\text{layer}} d_{\text{model}}^2$, where $n_{\text{layer}}$ is the number of layers and $d_{\text{model}}$ is the hidden dimension. For the open-source models Deepseek, Qwen, and DeepseekR1 (which is distilled from Qwen), these two hyperparameters come from [20, 25]. On the other hand, the information of the GPT models is unknown. We treat GPT-4o mini an 8B model and GPT-4o a 200B model, following [1]. We also treat OpenAI o3 the same size as GPT-4o, given no community information. Then, we estimate $n_{\text{layer}}$ and $d_{\text{model}}$ by borrowing the information from the LLaMA models of similar sizes [18]. Specifically, we reuse the configuration of LLaMA 8B for GPT-4o mini and that of LLaMA 405B for GPT-4o and OpenAI o3, except that for the latter case $n_{\text{layer}}$ is cut by half. The estimated FLOPS are given in Table 8.

Complementary to Figure 4 that compares different models/methods based on monetary costs and inference latencies, in Table 9, we list the number of LLM calls and the average number of tokens for solving one problem. The results are in three tiers. Qwen14B, GPT-4o mini, GPT-4o, and CoT make only one LLM call. The number of tokens per call is around 1000, with CoT costing slightly more because of its "thinking step by step" nature. DeepseekR1-14B and OpenAI o3 are reasoning models

Table 8: (Estimated) Model configurations and FLOPS per token.

| Model | $n_{\text{layer}}$ | $d_{\text{model}}$ | FLOPS |
|---|---|---|---|
| DeepSeek7B | 32 | 4096 | 12884901888 |
| Qwen7B | 28 | 3584 | 8631877632 |
| Qwen14B | 48 | 5120 | 30198988800 |
| GPT-4o mini | 32 | 4096 | 12884901888 |
| GPT-4o | 63 | 16384 | 405874409472 |
| DeepseekR1-14B | 48 | 5120 | 30198988800 |
| OpenAI o3 | 63 | 16384 | 405874409472 |

Table 9: LLM calls and token consumption per problem.

| | #LLM calls | #Tokens / call | #Tokens |
|---|---|---|---|
| Qwen14B | 1 | 949.50 | 949.50 |
| GPT-4o mini | 1 | 786.90 | 786.90 |
| GPT-4o | 1 | 810.25 | 810.25 |
| DeepseekR1-14B | 1 | 2210.15 | 2210.15 |
| OpenAI o3 | 1 | 1992.42 | 1992.41 |
| CoT | 1 | 1013.33 | 1013.33 |
| Reflexion | 20 | 1263.25 | 25265 |
| MapCoder | 16 | 2137.25 | 34196 |
| MoA | 10 | 957.90 + 1087.43 | 20453 |
| LessonL | 24 | 970.38 | 23289 |

and they cost more tokens per call (around 2000). Reflexion, MapCoder, MoA, and LessonL all solve a problem in many rounds, making more LLM calls and costing considerably more tokens. Among these four methods, LessonL requires the most number of calls but the per-call tokens are relatively economic, thanks to the concise design of lessons.

## J  Additional Experiment Results

### J.1  Complementary Strengths of Agents

As shown in Table 5, the models Deepseek7B, Qwen7B, and Qwen14B have complementary (but unknown a priori) strengths. Here, we study how much LessonL improves over the best of them category by category. Table 10 lists the speedup results.

We see that LessonL improves the speedup on many categories. Among them, "fft" and "stencil" enjoy the most significant improvement. Meanwhile, for categories that LessonL suffers from degradation of speedup, the degradation is only minor. In theory, LessonL is no worse than the best single agent. The degradation reflects randomness in LLM generation.

To investigate the complementary strengths of the agents, let us look into the "sort" category, on which Qwen7B performs poorly but Deepseek7B and Qwen14B are quite good. Without LessonL, Qwen7B fails to offer any speedup; with LessonL, it achieves $2.35\times$ speedup on average.

For a concrete example, consider Problem 40, sorting an array of complex numbers by magnitude. In this example, Qwen7B produces slower code initially but Deepseek7B and Qwen14B produce fast codes. These codes generate the following lessons:

> **Lesson from Deepseek7B**
>
> Code B is faster because it uses the 'std::norm' function, which calculates the square of the magnitude of a complex number, instead of the 'std::abs' function which calculates the true

Table 10: Performance comparison between the best single agent and LessonL across categories. Benchmark: ParEval (serial mode) speedup.

| Category | Best single agent | LessonL | Relative improvement |
|---|---|---|---|
| graph | 1.00 | 1.04 | 4.00% |
| histogram | 1.57 | 1.67 | 6.37% |
| scan | 5.95 | 5.91 | −0.67% |
| transform | 1.01 | 1.03 | 1.98% |
| sparse_la | 1.70 | 1.61 | −5.29% |
| reduce | 2.21 | 2.03 | −8.14% |
| fft | 1.02 | 2.23 | 118.63% |
| geometry | 8.09 | 9.80 | 21.14% |
| stencil | 1.12 | 3.23 | 188.39% |
| dense_la | 1.02 | 1.19 | 16.67% |
| sort | 2.35 | 2.69 | 14.47% |
| search | 1.07 | 1.02 | −4.67% |

> magnitude. Squaring the magnitude is faster and more efficient than calculating the square root, especially for large numbers.
>
> **Lesson from Qwen14B**
>
> Code B is significantly faster because it avoids recalculating the magnitude of each complex number multiple times during the sorting process by first storing all magnitudes in a separate vector. This reduces redundant computations and leverages efficient sorting algorithms on precomputed data.

These two lessons point to different optimizations: (1) using 'std::norm' rather than 'std::abs' to avoid a square root; (2) performing precomputation to avoid repetitively computing 'std::abs' inside the comparison function used for sorting. When given these lessons, Qwen7B generates code that achieves a speedup by following the guidance provided by the first lesson. When Qwen7B is not given any lessons, it fails to optimize this problem; it often generates code that fails to compile or attempts misguided optimizations like replacing 'std::abs' with two calls to 'std::hypot' (slower).

## J.2 Running LessonL for More Rounds

In addition to Figure 3 in the main text, we increased the interaction rounds of LessonL and allowed the agents to interact longer. Table 11 lists the speedups. It shows that the speedup improves when more computational resources are spent. However, there is a point after which the speedup nearly plateaus. This happens around round 8, indicating diminishing return afterward.

Table 11: Speedup over rounds for LessonL. Benchmark: ParEval (serial mode).

| Round | 1 | 2 | 3 | 4 | 5 | 6 | 7 | 8 | 9 | 10 |
|---|---|---|---|---|---|---|---|---|---|---|
| Speedup | 1.90 | 1.95 | 2.01 | 2.05 | 2.12 | 2.13 | 2.13 | 2.21 | 2.22 | 2.22 |

## K Case Study: Geometry

Let us study a case about finding the smallest distance among $n$ points in two-dimensional space. The original code performs a naive computation by enumerating all possible pairs and finding the minimum distance, which takes $O(n^2)$ time. The final optimized code reduces the complexity from $O(n^2)$ to $O(n \log n)$ and achieves an outstanding speedup of 74.31x. The algorithm starts by sorting the points based on their x-coordinates and then performs divide-and-conquer. The divide part separates a point set in two equal halves. The conquer part (recursively) calculates the minimum

distance for each of the two halves; let us call the smaller of the two distances, $d$. Then, the combine part filters out points with an x-distance from the middle point greater than $d$, sorts the remaining on their y-coordinates, ignores point pairs with a y-distance greater than $d$, and returns the smallest distance among the remaining points.

Qwen14B summarizes the reason of the gain as the following:

> *"Code B runs significantly faster than Code A because it sorts the points by their x-coordinate before comparing them pairwise. This sorting operation reduces the number of comparisons made in the nested loop, which is a time-consuming operation in Code A. By sorting the points first, we can eliminate unnecessary comparisons and thus speed up the execution time."*

We see that it attributes the initial sorting operation to be the main reason of the outstanding result. This reason is partially correct, but sorting is only the precursor step of divide-and-conquer that leads to a reduced $O(n \log n)$ cost. We proceed to investigate all the generated lessons over the course and find that indeed some lessons highlight the importance of divide-and-conquer.

Sorting is mentioned in lessons 1, 2, and 4, while divide-and-conquer is mentioned in lessons 1, 4, and 5. The remaining lessons are all about compilation failures, including forgetting '#pragma once' (lesson 3), no implementation of 'std::set' for 'Point' (lesson 6), mismatch of argument types (lesson 7), undefined functions (lesson 8), and use of function before definition (lesson 9). These lessons reflect the complex implementation of a divide-and-conquer algorithm, which can cause many errors (even for human programmers). Overall, the lessons are helpful to guide an LLM to generate correct and high-performing codes.

The original code, the final code, and the full text of the lessons are provided below. For readability, the codes were edited by removing nonessential comments (such as problem description) and white spaces (including indentation).

---

**Original code**

```cpp
#pragma once
#include <vector>
#include <limits>

struct Point {
  double x, y;
};

double distance(Point const& p1, Point const& p2) {
  return std::sqrt(std::pow(p2.x-p1.x, 2) + std::pow(p2.y-p1.y, 2));
}

/* Return the distance between the closest two points in the vector
points.
   Example:
   input: [{2, 3}, {12, 30}, {40, 50}, {5, 1}, {12, 10}, {3, 4}]
   output: 1.41421
*/
double closestPair(std::vector<Point> const& points) {
  // The polygon needs to have at least two points
  if (points.size() < 2)   {
    return 0;
  }
  auto getDist = [](Point const& a, Point const& b) {
    return std::sqrt(std::pow(b.x-a.x, 2) + std::pow(b.y-a.y, 2));
  };
  double minDist = std::numeric_limits<double>::max();
  for (size_t i = 0; i < points.size() - 1; i++) {
    for (size_t j = i + 1; j < points.size(); j++) {
```

```
      const double dist = getDist(points[i], points[j]);
      if (dist < minDist) {
        minDist = dist;
      }
    }
  }
  return minDist;
}
```

**Optimized code, speedup 74.31x**

```
#include <vector>
#include <algorithm>
#include <cmath>
#include <limits>

struct Point {
  double x, y;
};

double distance(const Point& p1, const Point& p2) {
  return std::sqrt(std::pow(p2.x - p1.x, 2) + std::pow(p2.y - p1.y,
  2));
}

double closestPairHelper(const std::vector<Point>& points, size_t
start, size_t end) {
  if (end - start <= 3) {
    double minDist = std::numeric_limits<double>::max();
    for (size_t i = start; i < end; ++i) {
      for (size_t j = i + 1; j < end; ++j) {
        minDist = std::min(minDist, distance(points[i], points[j]));
      }
    }
    return minDist;
  }

  size_t mid = (start + end) / 2;
  double leftMin = closestPairHelper(points, start, mid);
  double rightMin = closestPairHelper(points, mid, end);
  double minDist = std::min(leftMin, rightMin);

  std::vector<Point> strip;
  for (size_t i = start; i < end; ++i) {
    if (std::abs(points[i].x - points[mid].x) < minDist) {
      strip.push_back(points[i]);
    }
  }

  std::sort(strip.begin(), strip.end(), [](const Point& a, const
  Point& b) {
    return a.y < b.y;
  });

  for (size_t i = 0; i < strip.size(); ++i) {
    for (size_t j = i + 1; j < strip.size() && strip[j].y - strip[i].y
    < minDist; ++j) {
      minDist = std::min(minDist, distance(strip[i], strip[j]));
    }
  }
```

```
    return minDist;
}

double closestPair(const std::vector<Point>& points) {
  if (points.size() < 2) {
    return 0;
  }
  std::vector<Point> sortedPoints = points;
  std::sort(sortedPoints.begin(), sortedPoints.end(), [](const Point&
  a, const Point& b) {
    return a.x < b.x;
  });
  return closestPairHelper(sortedPoints, 0, sortedPoints.size());
}
```

**Lesson 1 (selected)**

Code B is significantly faster because it sorts the points by their x-coordinate before comparing each pair of points. This sorting step reduces the number of comparisons made in the nested loop, which is a time-consuming operation. By sorting the points first, we ensure that similar pairs of points (those with similar x-coordinates) are closer together in the sorted array, reducing the number of unnecessary distance calculations. This results in a significant speedup because it eliminates redundant computations.

**Lesson 2 (selected)**

Code B is significantly faster because it implements a divide-and-conquer strategy, also known as the "plane sweep" algorithm, which is more efficient for this type of problem. It sorts the points based on their x-coordinates and then uses a strip-based approach to find the closest pair of points. This approach reduces the time complexity from $O(n^2)$ to $O(n \log n)$, making it much faster for large datasets.

In Code A, the algorithm compares all pairs of points, leading to a time complexity of $O(n^2)$. This makes it less efficient for larger datasets compared to Code B, which has a better time complexity.

Additionally, Code B optimizes by only considering points within a certain distance (the minimum distance found so far) in the strip, reducing unnecessary comparisons. This optimization is not present in Code A and contributes to its slower performance.

**Lesson 3 (selected)**

Code B cannot be compiled because it is missing the '#pragma once' directive at the beginning of the file, which is used to prevent multiple inclusions of the same header file. Additionally, there are no syntactic errors in Code B that would cause it to fail compilation based on the provided compiler output.

**Lesson 4 (selected)**

Code B is significantly faster because it implements a more efficient algorithm for finding the closest pair of points. Specifically, it uses a divide-and-conquer strategy combined with the concept of a "strip" to reduce the time complexity from $O(n^2)$ to $O(n \log n)$.

In Code A, the algorithm compares every pair of points, which leads to a time complexity of $O(n^2)$. This is because there are $n * (n-1)/2$ comparisons in the worst case.

On the other hand, Code B uses a divide-and-conquer approach. It first sorts the points by their x-coordinate and then recursively finds the minimum distance in each half. This step has a time complexity of $O(n \log n)$.

Additionally, Code B also considers pairs of points that lie within a "strip" (a range of y-coordinates) around the vertical line that divides the two halves. This strip is sorted by y-coordinate and then checked for pairs that are closer than the current minimum distance. The stripClosest function has a time complexity of $O(n)$, which is used when there are more than 3 points in the strip.

### Lesson 5

Code B is significantly faster because it implements the divide-and-conquer algorithm for finding the closest pair of points, which reduces the time complexity from $O(n^2)$ in Code A to approximately $O(n \log n)$.

### Lesson 6

Code B cannot be compiled because it attempts to use a 'std::set' to store 'Point' objects, but 'Point' does not have a defined comparison operator. The compiler needs a way to compare 'Point' objects to insert them into the set.

### Lesson 7

Code B cannot be compiled because the 'stripClosest' function is called with a 'std::vector<Point>' and a 'double', but the function expects a 'std::vector<Point>' and an 'int'. The compiler output would indicate a type mismatch error at the call to 'stripClosest'.

### Lesson 8

Code B cannot be compiled because the function 'closestPairUtil' is declared but not defined, leading to a linker error. The compiler output would indicate an unresolved external symbol for 'closestPairUtil'.

### Lesson 9

Code B cannot be compiled because the 'stripClosest' function is defined after it is used in the 'closestPairUtil' function. In C++, functions must be declared before they are used, or defined inline within the scope where they are called. The compiler error indicates that it does not recognize 'stripClosest' when it is first referenced in 'closestPairUtil'.

## L    Case Study: DFT

Let us study a case about performing the discrete Fourier transform (DFT). Mathematically, the DFT of a length-$N$ array $x$ is an array $X$ of the same length, where

$$X_k = \sum_{n=0}^{N-1} x_n \cdot e^{-i2\pi nk/N} = \sum_{n=0}^{N-1} x_n[\cos(-2\pi nk/N) + \sin(-2\pi nk/N)], \quad k = 0, 1, \ldots, N-1.$$
(1)

Because of the periodicity of trigonometric functions, many $nk$ values are congruent modulo $N$ and thus many complex exponentials are repeatedly computed. A natural idea of improving the runtime of the DFT is to precompute these exponentials and perform a table lookup whenever needed. Indeed, the optimized code adopts this idea, reflected by lessons 1, 2, 5, 6, and 8, and agreed with by Qwen14B's assessment:

> *"Code B is significantly faster than Code A because it precomputes the complex exponential factors, reducing redundant calculations in the inner loop. This*

> *optimization minimizes the computational overhead and accelerates the DFT process."*

It is interesting to note that the above mathematical definition of DFT implies an $O(N^2)$ computational cost, while it is well known that a divide-and-conquer algorithm, called the fast Fourier transform (FFT), can bring down the cost to $O(N \log N)$. Some agents in some rounds do try FFT, as lessons 4, 7, and 9 suggest, but they fail. Two of these lessons say that the optimized code "is nonequivalent to" the original code, while another lesson says there is a logical error. We investigate into the resulting codes and find that they make an error by flipping the sign (note the negative sign inside the trigonometric functions in (1)). In other words, the incorrectness of the codes is caused by a sign error, rather than the nonequivalence between DFT and FFT. While the lessons are not fully correct, they do discourage the agents to try the FFT idea, as the agents are probably not capable of precisely implementing FFT in context.

Another lesson (lesson 3) is incurred from compilation errors. The code attempts to use Intel intrinsic instructions but it fails compilation. Such attempts requires CPU support and the use of the corresponding compiler flags, which are not supported by our evaluation pipeline. This lesson discourages the use of Intel intrinsics in the benchmarking.

The original code, the final code, and the full text of the lessons are provided below. For readability, the codes were edited by removing nonessential comments (such as problem description) and white spaces (including indentation).

---

**Original code**

```cpp
#pragma once
#include <vector>
#include <complex>
#include <cmath>

#if !defined(M_PI)
#define M_PI 3.14159265358979323846
#endif
/* Compute the discrete fourier transform of x. Store the result in
output.
   Example:
   input: [1, 4, 9, 16]
   output: [30+0i, -8-12i, -10-0i, -8+12i]
*/
void dft(std::vector<double> const& x,
std::vector<std::complex<double>> &output) {
  int N = x.size();
  output.resize(N, std::complex<double>(0, 0)); // Resize the output
  vector and initialize with 0
  for (int k = 0; k < N; k++) { // For each output element
    std::complex<double> sum(0, 0);
    for (int n = 0; n < N; n++) { // For each input element
      double angle = 2 * M_PI * n * k / N;
      std::complex<double> c(std::cos(angle), -std::sin(angle)); //
      Euler's formula
      sum += x[n] * c;
    }
    output[k] = sum;
  }
}
```

**Optimized code, speedup 10.83x**

```cpp
#pragma once
```

```cpp
#include <vector>
#include <complex>
#include <cmath>

#if !defined(M_PI)
#define M_PI 3.14159265358979323846
#endif

void dft(std::vector<double> const& x,
std::vector<std::complex<double>> &output) {
  int N = x.size();
  output.resize(N, std::complex<double>(0, 0)); // Resize the output
  vector and initialize with 0

  // Precompute the complex exponential factors
  std::vector<std::complex<double>> W(N);
  for (int k = 0; k < N; k++) {
    double angle = -2 * M_PI * k / N;
    W[k] = std::complex<double>(std::cos(angle), std::sin(angle));
  }

  for (int k = 0; k < N; k++) { // For each output element
    std::complex<double> sum(0, 0);
    for (int n = 0; n < N; n++) { // For each input element
      sum += x[n] * W[(n * k) % N];
    }
    output[k] = sum;
  }
}
```

**Lesson 1 (selected)**

Code B is significantly faster because it precomputes the complex exponential factors and stores them in a vector, W. This avoids the need to compute the same factor multiple times within the nested loop, which can lead to significant performance improvements due to reduced computation time and increased cache efficiency. In Code B, each factor is computed only once and then reused in the inner loop, while in Code A, each factor is recomputed for every input element.

**Lesson 2 (selected)**

Code B is significantly faster than Code A because it precomputes the complex exponential factors, reducing redundant calculations in the inner loop. This optimization minimizes the overhead associated with trigonometric function calls and repeated angle calculations, leading to a substantial performance improvement.

**Lesson 3 (selected)**

The compiler output is not provided, but based on the code, the issue likely lies in the use of '_mm256_set_pd' and '_mm256_mul_pd' with 'double' values. These intrinsics expect '__m256d' (256-bit vector of 4 double-precision floating-point numbers), but the code is passing individual 'double' values. Additionally, the horizontal add operations are incorrectly used to sum up the results.

**Lesson 4 (selected)**

Code B is nonequivalent to Code A because it uses the Cooley-Tukey FFT algorithm, which is a fast Fourier transform (FFT) method, while Code A implements the direct computation of the Discrete Fourier Transform (DFT). The FFT algorithm in Code B is more efficient for large input sizes, but it does not handle real-valued inputs directly; instead, it processes complex numbers. Code A correctly handles real-valued inputs by converting them to complex numbers with zero imaginary parts before computing the DFT.

**Lesson 5**

Code B is faster because it precomputes the complex exponential factors (W[k]) and reuses them in the inner loop, avoiding redundant calculations. In Code A, the complex exponential factor is calculated inside the inner loop for each element of x, which results in a higher number of operations and thus slower execution time.

**Lesson 6**

Code B is significantly faster because it precomputes the complex exponential factors (W[n]) and reuses them in the inner loop, which reduces the number of trigonometric function calls and thus improves performance. In Code A, each iteration of the inner loop computes a new complex exponential, while in Code B, these values are precomputed and stored in the W vector, leading to a more efficient use of computational resources.

**Lesson 7**

Code B is not equivalent to Code A because it uses the Fast Fourier Transform (FFT) algorithm instead of the direct computation of the Discrete Fourier Transform (DFT). The FFT is an efficient algorithm for computing the DFT, but it requires the input size to be a power of two. If the input size is not a power of two, Code B will still attempt to compute the FFT, which can lead to incorrect results or errors. In contrast, Code A directly computes the DFT using nested loops, which does not have this restriction but may be less efficient for large input sizes.

**Lesson 8**

Code B is significantly faster than Code A because it precomputes the complex exponential factors, reducing redundant calculations in the inner loop. This optimization minimizes the overhead of trigonometric function calls and repeated angle calculations, leading to a substantial performance improvement.

**Lesson 9**

Code B contains a logical error in the recursive call to 'dft' within the Cooley-Tukey FFT algorithm. The output vector is being modified in-place, and the recursive calls are overwriting parts of the output vector that are still needed for the computation. This leads to incorrect results and potential crashes.

# M   Case Study: Sum of Prefix Sum (A Failure Example)

Some problems in the benchmarks appear challenging for small LLMs, even if a few of them collaborate. Here, we study one problem from ParEval (`32_scan_sum_of_prefix_sum_array`), which asks to write efficient code to compute the sum of a prefix sum.

In the initial round, none of the agents writes compilable codes. These codes are buggy for different reasons, either missing headers or placing '#pragma once' in the wrong place, as the following lessons cite.

Code B cannot be compiled because the '#pragma once' directive is incorrectly placed. It should be at the beginning of the file, before any other code or comments. Additionally, the compiler output would indicate syntax errors or warnings related to the placement of '#pragma once'.

Subsequently, all agents try to fix syntax errors but other errors occur. Another common error is that the output code computes the sum rather than the sum of the prefix sum, such as the following:

```cpp
double sumOfPrefixSum(std::vector<double> const& x) {
  double prefixSum = 0.0;
  for(double num : x) {
    prefixSum += num;
  }
  return prefixSum;
}
```

Occasionally, an agent hallucinates some code that resembles the formula $n \times (n+1)/2$ for computing the sum of consecutive integers from $1$ to $n$, but this formula is inapplicable to the current problem and thus the code is incorrect.

```cpp
double sumOfPrefixSum(std::vector<double> const& x) {
  double sum = 0.0;
  for (double num : x) {
    sum += num;
  }
  return sum * (x.size() + 1) / 2;
}
```

Sometimes the agents do write correct codes, but yield no speedup:

```cpp
double sumOfPrefixSum(const std::vector<double>& x) {
  double prefixSum = 0.0;
  double totalSum = 0.0;
  for (double value : x) {
    prefixSum += value;
    totalSum += prefixSum;
```

```
    }
    return totalSum;
}
```

This example involves various failures an LLM could produce: syntax errors, semantic errors, and lack of speedup. One mitigation is to integrate retrieval-based methods from external data sources to provide relevant code examples and optimization strategies, helping reduce syntatic errors and brainstorm optimization methods. This will require building comprehensive knowledge bases and high-quality code libraries.

## N   Supporting Code

Code implementation is available at `https://github.com/MITIBM-FastCoder/LessonL`.

