# OpenReview forum: "Lessons Learned: A Multi-Agent Framework for Code LLMs to Learn and Improve"
_NeurIPS.cc/2025/Conference — NeurIPS 2025 poster_

### Official Review · Reviewer_r8DA · 2025-06-29

**Clarity:** 3
**Significance:** 2
**Originality:** 2
**Rating:** 3
**Confidence:** 4

**Summary:**

This paper presents LessonL, a multi-agent collaboration framework where multiple LLM agents iteratively improve their code optimization and generation capabilities by learning from each other's successes and failures. The core mechanism involves three components: (1) lesson solicitation, where agents generate explanatory lessons from their solution attempts; (2) lesson banking and selection, where lessons are stored and filtered based on effectiveness and relevance; and (3) effectiveness adjustment, which dynamically updates lesson utility scores. The authors evaluate their approach on code optimization (ParEval, PolyBench) and generation (HumanEval, MBPP) benchmarks, demonstrating that small model ensembles can outperform larger individual models under similar resource constraints.

**Questions:**

1. Can the lessons accumulated in this way be used to train LLMs and make them more powerful?
2. How does performance scale with increasing numbers of agents (n > 3) and interaction rounds (T > 4)?
3. Could you provide statistics on the average number of LLM calls per method across different benchmarks? LessonL requires additional lesson solicitation steps beyond the main solution generation, and with multiple rounds and agents, the total LLM invocations could be significantly higher than baseline methods. A detailed breakdown showing  for each method would provide better understanding of the true computational overhead and help validate the cost-effectiveness claims in Figure 4.

**Ethical Concerns:**

["NO or VERY MINOR ethics concerns only"]

**Final Justification:**

The author's rebuttal addresses some of my concerns. However, the key issue regarding the heuristic lesson selection mechanism remains unconvincing, particularly as coding tasks become more complex. In real-world, project-level scenarios, the current approach of selecting lessons based on a combination of speedup and semantic scores from retrieved code patches seems overly simplistic. Even at a small scale, this method incurs high token costs, which would likely be even more problematic at larger scales. From a human learning perspective, we don't merely accumulate code patches; rather, we abstract higher-level error patterns and formulate coding rules to guide future improvements. While the current lesson mechanism shows promise for small-scale tasks, its generalizability and applicability to realistic, large-scale coding projects remain limited. Therefore, I will maintain my original score.

**Limitations:**

Yes

**Paper Formatting Concerns:**

The pseudocode at Line 108-115 could use algorithm format instead of plain text.

**Quality:**

2

**Strengths And Weaknesses:**

### Strengths

- The framework achieves substantial and consistent improvements across different metrics and benchmarks, with some cases showing dramatic performance gains.
- The paper provides extensive experiments across multiple benchmarks for both code optimization and generation, with proper statistical analysis including error bars and multiple runs across different programming modes.
- This paper reveals that different LLMs have their own areas of expertise in code optimization tasks, and their capabilities can be enhanced by learning from lessons (accumulating experience).

### Weaknesses

1. The core contribution seems limited as multi-agent collaboration with experience sharing has been extensively explored in existing collaborative learning frameworks[1], with the main difference being application to coding tasks.
2. The lesson selection mechanism (splitting selections half-and-half between high-speedup and high-relevance) appears arbitrary. The effectiveness adjustment using simple linear interpolation with epsilon parameter lacks sophistication and theoretical grounding. Could you explain more about this selection mechanism qualitatively and quantitively?
3.  The introduction motivates the work by highlighting that different LLMs have complementary strengths in different coding categories (e.g., Qwen7B excels at "geometry" while Deepseek7B excels at "scan"). However, the proposed framework has LLMs learn from the same LLMs than leveraging different LLMs' strengths. This creates a logical inconsistency. Could you provide more supportive experiments to support this claim?

[1] Li J, Lai Y, Li W, et al. Agent hospital: A simulacrum of hospital with evolvable medical agents[J]. arXiv preprint arXiv:2405.02957, 2024.

---

> ### Author Rebuttal · Authors · 2025-07-31
>
> Thank you for the thoughtful feedback. We appreciate your recognition of the robustness of our results, the comprehensiveness of our evaluation, and the insight into how diverse LLM expertise can be amplified through shared learning.
>
> We will adjust the pseudocode.
>
> ### W1: The core contribution seems limited.
>
> We appreciate the pointer to [1], Agent Hospital, which is an overlooked related work we will cite. However, our work is more than applying an existing idea to a new application. Our lessons are analogous to experiences, but a main difference of our method from Agent Hospital is how the agents collaborate and consequently how the lessons/experiences are shared.
>
> In Agent Hospital, medical agents treat a large number of patients and accumulate corresponding case information as well experiences. Later, an agent can retrieve past experiences and similar cases to assist treating new patients. It is not mentioned whether several doctor agents collaboratively treat the same patient, but it's likely that agents with different roles (such as a doctor agent and a nurse agent) work together. In this case, the collaboration of the agents are role-based and they don't seem to share experiences with each other; rather, they retrieve experiences from the knowledgebase.
>
> In contrast, our coding agents collaboratively solve one coding task by sharing their experiences/lessons on this task. Hence, lesson sharing poses a few questions unanswered by Agent Hospital: (1) will collaboration with lesson sharing be better than non-collaboration or collaboration without lesson sharing; (2) do we need to build the lesson bank over tasks or are the lessons for the current task sufficiently helpful. The work of Agent Hospital cannot answer question (1) because doctor agents cannot work without nurse agents and they don't share experiences in treating the same patient. For question (2), Agent Hospital builds experiences over the treatment of many patients (i.e., many problems), while we share lessons in an individual problem. We show that even lessons not accumulated over problems are quite helpful.
>
> ### W2: The lesson selection mechanism appears arbitrary.
>
> Thank you for the comment. Qualitatively, the retrieval of k/2 lessons based on similarity / relevance serves two practical purposes in our work: (a) A lesson with low speedup might still be helpful if it helps other agents avoid a common mistake/pitfall; (b) semantically similar lessons tend to be more actionable as they often reference specific functions / patterns in the code. In practice, a 50/50 split works well.
>
> The speedup $s$ associated with a lesson is based on the performance of a single program generated by a single agent. In order to weigh lessons based on their ability to help *all* collaborating agents optimize code, we introduce the adjustment factor $f$. Intuitively, the purpose of the adjustment factor is to reward lessons that "lift-up" other collaborating agent's performance, and decay the weight of lessons that do not.
>
> The details of how the adjustment factor $f$ is updated involves a heuristic that is described in section 3.3 of the paper and briefly summarized here. We aggregate $1\pm\epsilon$ terms over all agents based on whether the agent's performance is above or below the target when using the lesson and then compute the average. Note that speedup due to code optimization can be highly variable across different agents and different problem categories. Our simple approach avoids potential pitfalls in which a single agent's performance disproportionately influences the weight of a lesson. Although simple, the scheme is empirically effective as illustrated in the paper's ablation study.
>
> ### W3: The introduction motivates the work by highlighting that different LLMs have complementary strengths in different coding categories. Could you provide more supportive experiments to support this claim?
>
> Thank you for raising this topic. We can indeed provide some additional supporting evidence for this claim.
>
> Consider the performance of Qwen7B on the "Sort" category of ParEval (shown in Table 4 in the appendix), where Deepseek7B and Qwen14B are quite good. Without LessonL Qwen7B achieves no speedup (speedup 1.0) on the "Sort" problems. When given lessons from the last round of LessonL, however, Qwen7B achieves 2.63x speedup on the "Sort" problems.
>
> Let us briefly look at a concrete example from the "sort" category. The example is problem 40, sorting an array of complex numbers by magnitude. In this example, Qwen7B produces slower code initially but Deepseek7B and Qwen14B produce fast codes. These codes generate the following lessons:
>
> > (Deepseek7B) Code B is faster because it uses the std::norm function, which calculates the square of the magnitude of a complex number, instead of the std::abs function which calculates the true magnitude. Squaring the magnitude is faster and more efficient than calculating the square root, especially for large numbers.
>
> > (Qwen14B) Code B is significantly faster because it avoids recalculating the magnitude of each complex number multiple times during the sorting process by first storing all magnitudes in a separate vector. This reduces redundant computations and leverages efficient sorting algorithms on precomputed data.
>
> These two lessons point to different optimizations: (1) using std::norm rather than std::abs to avoid a square root; (2) performing precomputation to avoid repetitively computing std::abs inside the comparison function used for sorting. When given these lessons, Qwen7B generates code that achieves a $21.3\times$ speedup by following the guidance provided by the first lesson. When Qwen7B is not given any lessons, it fails to optimize this problem; it often generates code that fails to compile or attempts misguided optimizations like replacing `std::abs` with two calls to `std::hypot` (slower).
>
> ### Q1: Can the lessons accumulated in this way be used to train LLMs and make them more powerful?
>
> We believe that the generated lessons are useful assets and they can be reused in different ways to improve LLMs. For example, one may use the lessons to fine-tune reasoning models. Specifically, one may construct a chain of increasingly faster codes by pruning out slow/non-compilable codes and repositioning the lessons from post-code analysis to pre-code reasoning. This produces training traces that can enhance the reasoning ability of code LLMs.
>
> ### Q2: How does performance scale with increasing numbers of agents (n > 3) and interaction rounds (T > 4)?
>
> We conduct additional experiments by increasing n and T. Overall, doing so can improve the correctness and speedup but requires proportionally more resources. The return is diminishing after some point. Here are the details.
>
> **Increasing the number of agents.** We add three more agents to the experiment: Llama-3.1-8B-Instruct, Qwen2.5-7B-Instruct, and phi-4 (14B). These models have a similar size to the three already used in the paper submission (Deepseek7B, Qwen7B, Qwen14B). We repeat the set up of Table 1 from the paper for ParEval serial and obtain the following results.
>
> | # Agents | Correct | >2× | Speedup |
> |---------:|--------:|----:|--------:|
> | 1 | 0.67 | 0.14 | 1.60 |
> | 3 | 0.90 | 0.23 | 2.12 |
> | 4 | 0.93 | 0.22 | 2.14 |
> | 5 | 0.93 | 0.20 | 2.09 |
> | 6 | 0.95 | 0.23 | 2.21 |
>
> The results suggest that generally, using more agents lead to better performance. This observation is uniform for all metrics, including correctness, the proportion of problems with >2x speedup, and the overall speedup. However, the greatest improvement comes from using three agents; using more generates diminishing returns.
>
> **Increasing the interaction rounds.** We repeat the set up of Figure 3 from the paper by allowing the agents to interact longer. The results are summarized in the following table.
>
> | Round | Speedup |
> |------:|----------------:|
> | 1 | 1.90 |
> | 2 | 1.95 |
> | 3 | 2.01 |
> | 4 | 2.05 |
> | 5 | 2.12 |
> | 6 | 2.13 |
> | 7 | 2.13 |
> | 8 | 2.21 |
> | 9 | 2.22 |
> | 10 | 2.22 |
>
> Like in the previous case, the speedup improves when more computational resources are spent. However, if one visualizes the speedup curve, one will observe that there is a point after which the speedup nearly plateaus. This happens around round 8, which goes beyond the five rounds experimented in the paper submission.
>
> ### Q3: Could you provide statistics on the average number of LLM calls per method across different benchmarks?
>
> In the following table, we list the number of LLM calls for each method, together with the number of tokens per call and the total number of tokens for solving one problem. This table is complementary to Figure 4, which additionally takes into account the monetary costs and latencies of different LLMs.
>
> | Baseline    | LLM Calls | Tokens / Call | Tokens |
> |-------------|-----------|---------------|--------|
> | Qwen14B     | 1         | 949.5 | 949.5 |
> | GPT-4o mini | 1         | 786.9 | 786.9 |
> | GPT-4o      | 1         | 810.25 | 810.25 |
> | CoT         | 1         | 1013.33 | 1013.33 |
> | Reflexion   | 20        | 1263.25 | 25265 |
> | MapCoder    | 16        | 2137.25 | 34196 |
> | MoA         | 10        | 957.9 + 1087.43 | 20453 |
> | LessonL     | 24        | 970.38 | 23289 |
>
> Considering only tokens, the first four and the rest are in drastically different tiers. The first four are only single LLMs without solving problems in rounds. Reflexion is also a single LLM but it solves problems in many rounds. MapCoder, MoA, and LessonL are multi-agent methods, which consume considerably more tokens and require more LLM calls. Among the last three, LessonL requires the most number of calls but the per-call tokens are relatively economic, thanks to the concise design of lessons.
>
> Thank you again for your constructive feedback.
>
> ### References
>
> [1] Li et al. Agent hospital: A simulacrum of hospital with evolvable medical agents. Preprint arXiv:2405.02957, 2024.

---

> > ### Comment · Reviewer_r8DA · 2025-08-05
> > **Response**
> >
> > Thanks for the rebuttal. The author's rebuttal addresses some of my concerns. However, the key issue regarding the heuristic lesson selection mechanism remains unconvincing, particularly as coding tasks become more complex. In real-world, project-level scenarios, the current approach of selecting lessons based on a combination of speedup and semantic scores from retrieved code patches seems overly simplistic. Even at a small scale, this method incurs high token costs, which would likely be even more problematic at larger scales. From a human learning perspective, we don't merely accumulate code patches; rather, we abstract higher-level error patterns and formulate coding rules to guide future improvements. While the current lesson mechanism shows promise for small-scale tasks, its generalizability and applicability to realistic, large-scale coding projects remain limited.

---

> > > ### Author Response · Authors · 2025-08-07
> > >
> > > Thank you for the comment. We're pleased to hear that some of the concerns have been addressed.
> > >
> > > For the issue of lesson selection, while we respect your concern that it is heuristic, we do want to add two points for a global consideration. First, the focus of this work is a framework for LLM agents to collaborate; lesson selection is a part of the framework. Inside this framework, the ablation study supports that doing selection is better than not and that alternative selection mechanisms are worse than what we propose. Second, there is flexibility in the selection criteria depending on the task. The heavily discussed criteria (speedup and semantic relevance) are tailored for code optimization. For other tasks, the criteria will need to be adapted accordingly. For example, for code generation, we replace the speedup criterion with the number of test cases passed (as discussed in Line 167 of the main text). For project-level scenarios like bug fix in SWE-Bench [1] (which we have not experimented with), the mechanism will need to be correspondingly more sophisticated. We speculate that a match of errors/failures to lessons in the pool can lead to more personalized and informative selection.
> > >
> > > > From a human learning perspective, we don't merely accumulate code patches; rather, we abstract higher-level error patterns and formulate coding rules to guide future improvements.
> > >
> > > We fully agree with your valuable insight. The concept of a lesson in LessonL is designed to reflect this perspective: instead of raw code patches, we let a lesson contain verbal descriptions and reasoning behind the change of the code. In code optimization, the lesson describes the transformation from the source to the current implementation and rationalizes why this change leads to the corresponding results. In project-level bug fix tasks, the lesson could similarly be a concise textual description and reasoning of the changes between working and non-working code, rather than a large code block or a diff between commits.
> > >
> > > Thank you again for engaging with us and providing thoughtful feedback. We hope this response clarifies misunderstandings, and we welcome any further questions or suggestions.
> > >
> > > [1] Jimenez, Carlos E., et al. "Swe-bench: Can language models resolve real-world github issues?." arXiv preprint arXiv:2310.06770 (2023).

---

### Official Review · Reviewer_TW55 · 2025-07-03

**Clarity:** 4
**Significance:** 3
**Originality:** 3
**Rating:** 5
**Confidence:** 4

**Summary:**

The paper introduces LessonL, a novel multi-agent framework for collaborative problem-solving in code optimization and generation. It leverages the complementary strengths of multiple LLM agents, enabling them to learn from each other's successes and failures. The framework features a lesson solicitation–banking–selection mechanism, facilitating iterative improvement of solutions. Empirical evaluations demonstrate that LessonL outperforms single-agent and multi-agent baselines, achieving state-of-the-art performance on benchmarks like ParEval and HumanEval. Ablation studies confirm the effectiveness of key components, and cost analysis shows LessonL's efficiency. Case studies highlight insightful optimizations, such as divide-and-conquer strategies and precomputation techniques.

**Questions:**

Could you provide some failed examples generated by LessonL and discuss their potential mitigation strategies?

This would help better understand the framework's limitations and areas for improvement.

**Ethical Concerns:**

["NO or VERY MINOR ethics concerns only"]

**Final Justification:**

all of the concerns are resolved.

**Limitations:**

yes

**Quality:**

4

**Strengths And Weaknesses:**

Strengths:

- A novel multi-agent framework for Code LLMs that enables collaborative learning and knowledge sharing between agents.

- Comprehensive experimental evaluation across multiple LLMs and agent architectures, demonstrating consistent performance improvements.

- Clear and detailed demonstration of the framework's effectiveness through ablation studies and case studies that highlight specific optimization strategies.


Weaknesses:

- The paper lacks sufficient justification for selecting the specific three open-source LLMs used in the experiments.

- The framework's generalizability from code optimization to code generation tasks appears limited, with relatively weaker performance in the latter domain. This limitation warrants further investigation.

---

> ### Author Rebuttal · Authors · 2025-07-31
>
> Thank you for the positive and constructive feedback. We’re grateful for your recognition of the framework’s novelty, the depth of our experimental evaluation, and the clarity of our case studies.
>
> Please find our responses to your comments below.
>
>
> ### W1: The paper lacks sufficient justification for selecting the specific three open-source LLMs used in the experiments.
>
>
> Our agent selection is driven by the goal of demonstrating that a team of small, accessible LLMs can outperform large LLMs under a comparable budget. To that end, we focus on open-source models around the 10B parameter scale—models that can be comfortably run on a commodity GPU or accessed through LLM providers with a low cost. Given our focus on coding tasks, we prioritize models that are pre-trained or fine-tuned on code. Beyond these constraints, our choice of models is otherwise arbitrary among those that meet the criteria.
>
>
> ### W2: The framework's generalizability from code optimization to code generation tasks appears limited, with relatively weaker performance in the latter domain. This limitation warrants further investigation.
>
>
> Code optimization is the main focus of this project, but we see that the concept of collaboration through learning from each other's lessons is ubiquitous, as it reflects human practice. Continuously learning from lessons substantially benfits the optimization task because in addition to writing code correctly (such as in code generation and bug fix), which has a one-off pass/failure assessment, the improvement of code speed does not have an obvious limit. Code performance can be greatly improved by a large diverse set of optimization strategies, with potential speedups exceeding 1000x. Even for code generation, our experiment results show that the proposed method outperforms strong baselines (i.e., our performance has a weaker improvement but is not weaker). Indeed, we agree with the reviewer that this paper invites further investigations and we do plan to expand the benchmarking to more coding tasks, such as SWE-bench.
>
>
> ### Q1: Could you provide some failed examples generated by LessonL and discuss their potential mitigation strategies?
>
>
> Thank you for the helpful question. Some problems in the benchmarks appear challenging for small LLMs, even if a few of them collaborate. Here, we study one problem from ParEval (32_scan_sum_of_prefix_sum_array), which asks to write efficient code to compute the sum of a prefix sum.
>
> In the initial round, none of the agents writes compilable codes. These codes are buggy for different reasons, either missing headers or placing `#pragma once` in the wrong place, as the following lessons cite:
>
> > (Deepseek7B) Code B cannot be compiled because it lacks the necessary header files and does not include them in its code. The function `std::inclusive_scan` and `std::accumulate` are part of the `<numeric>` header, which is not included in Code B. Similarly, the `std::vector` class is defined in the `<vector>` header, which is also missing from Code B.
>
> > (Qwen7B) Code B cannot be compiled because it lacks the necessary header files and namespace declaration for the standard library functions it uses. The `std::vector` and `std::accumulate` functions are part of the `<vector>` and `<numeric>` headers, respectively, but these are not included in Code B. Additionally, Code B needs to be within a function or class to compile successfully.
>
> > (Qwen14B) Code B cannot be compiled because the `#pragma once` directive is incorrectly placed. It should be at the beginning of the file, before any other code or comments. Additionally, the compiler output would indicate syntax errors or warnings related to the placement of `#pragma once`.
>
> Subsequently, all agents try to fix syntax errors but other errors occur. Another common error is that the output code computes the sum rather than the sum of the prefix sum, such as the following:
>
> ```
> double sumOfPrefixSum(std::vector<double> const& x) {
>     double prefixSum = 0.0;
>     for(double num : x) {
>         prefixSum += num;
>     }
>     return prefixSum;
> }
> ```
>
> Occassionally, an agent hallucinates some code that resembles the formula n * (n + 1) /2 for computing the sum of consecutive integers from 1 to n, but this formula is inapplicable to the current problem and thus the code is incorrect.
>
> ```
> double sumOfPrefixSum(std::vector<double> const& x) {
>     double sum = 0.0;
>     for (double num : x) {
>         sum += num;
>     }
>     return sum * (x.size() + 1) / 2;
> }
> ```
>
> Sometimes the agents do write correct codes, but yield no speedup:
>
> ```
> double sumOfPrefixSum(const std::vector<double>& x) {
>     double prefixSum = 0.0;
>     double totalSum = 0.0;
>     for (double value : x) {
>         prefixSum += value;
>         totalSum += prefixSum;
>     }
>     return totalSum;
> }
> ```
>
> This example involves various failures an LLM could produce: syntax errors, semantic errors, and lack of speedup. One mitigation is to integrate retrieval-based methods from external data sources to provide relevant code examples and optimization strategies, helping reduce syntatic errors and brainstorm optimization methods. This will require building comprehensive knowledgebases and high-quality code libraries.
>
> Thank you again for your feedback.

---

> > ### Comment · Reviewer_TW55 · 2025-08-03
> >
> > Thanks for your response!

---

### Official Review · Reviewer_xyGF · 2025-07-04

**Clarity:** 3
**Significance:** 3
**Originality:** 3
**Rating:** 4
**Confidence:** 3

**Summary:**

This paper introduces LessonL, a multi-agent framework that enables code LLMs to collaboratively solve coding problems through shared learning. The key innovation is a "lesson" mechanism—concise knowledge pieces that explain why solutions succeed or fail—allowing agents to learn from each other's experiences. Unlike existing multi-agent approaches that assign fixed roles or use simple voting, LessonL enables dynamic knowledge transfer among agents.
The framework operates through four iterative stages: (1) Lesson Solicitation from solution outcomes, (2) Lesson Banking in a central repository, (3) Lesson Selection based on speedup and semantic relevance, and (4) Solution Generation using selected lessons. The authors demonstrate that teams of small models (7B-14B parameters) can outperform larger models like GPT-4o on both code optimization and generation tasks across six benchmarks.

**Questions:**

How does performance scale with 5+ agents? Is there a point of diminishing returns or negative interference?
How do you handle potentially incorrect lessons? Have you observed cases where misleading lessons degraded performance across multiple agents?
How does LessonL compare with recent test-time compute scaling approaches like those used in DeepSeek-R1(-Distill)

**Ethical Concerns:**

["NO or VERY MINOR ethics concerns only"]

**Quality:**

3

**Strengths And Weaknesses:**

Strengths
The lesson-based collaboration mechanism draws natural inspiration from human peer learning. This represents a significant departure from fixed role-based (planner, coder, debugger) or voting-based multi-agent systems.

The paper evaluates on 6 benchmarks with thorough ablation studies validating each component. The cost-effectiveness analysis considering both monetary and computational costs is particularly valuable. Results demonstrate statistical rigor with multiple runs and standard deviations reported.

The paper aims to address code optimization—an important but understudied area compared to code generation. The interpretable lessons produced as byproducts have educational value.

Achieves state-of-the-art results, with impressive speedups and the surprising finding that small model teams can outperform GPT-4o.

Weaknesses
Scalability: The framework is only tested with 3 agents, leaving unclear how performance and coordination complexity scale with more agents. Evaluation is limited to 5 rounds, so long-term behavior is unknown. There's no comparison with recent test-time scaling methods (e.g., DeepSeek-R1-Distill/Qwen 3, o1-style reasoning) beyond basic CoT and Reflexion.

Lesson Quality and Generalization: The paper lacks mechanisms to verify lesson correctness beyond empirical performance, creating risk of error propagation through incorrect lessons. There's no discussion of handling conflicting or contradictory lessons. The paper doesn't evaluate cross-benchmark transfer—whether lessons from one benchmark help on another.

Analysis: Missing analysis of which types of problems benefit most from this approach.

---

> ### Author Rebuttal · Authors · 2025-07-31
>
> Thank you for your thoughtful review. We greatly appreciate your recognition of the novelty of our method, the value of the analysis, the statistical rigor of the experiments, and the joy of finding that small model teams can outperform GPT-4o.
>
> We address your comments in the following.
>
> ### W1. Scalability: The framework is only tested with 3 agents, leaving unclear how performance and coordination complexity scale with more agents. Evaluation is limited to 5 rounds, so long-term behavior is unknown. There's no comparison with recent test-time scaling methods (e.g., DeepSeek-R1-Distill/Qwen 3, o1-style reasoning) beyond basic CoT and Reflexion.
>
> We conduct additional experiments to respond to these points. Overall, we find that using more agents or running more rounds can improve the correctness and speedup but requires proportionally more resources. The return is diminishing after some point. Additionally, while not all reasoning models are competitive, o3 surprises us with either better correctness or better speedup (but not both). Nevertheless, one needs to pay nearly three times the money to enjoy the competitiveness of o3. Here are the details.
>
> **Increasing the number of agents.** We add three more agents to the experiment: Llama-3.1-8B-Instruct, Qwen2.5-7B-Instruct, and phi-4 (14B). These models have a similar size to the three already used in the paper submission (Deepseek7B, Qwen7B, Qwen14B). We repeat the set up of Table 1 from the paper for ParEval serial and obtain the following results.
>
> | # Agents | Correct | >2× | Speedup |
> |---------:|--------:|----:|--------:|
> | 1 | 0.67 | 0.14 | 1.60 |
> | 3 | 0.90 | 0.23 | 2.12 |
> | 4 | 0.93 | 0.22 | 2.14 |
> | 5 | 0.93 | 0.20 | 2.09 |
> | 6 | 0.95 | 0.23 | 2.21 |
>
> The results suggest that generally, using more agents lead to better performance. This observation is uniform for all metrics, including correctness, the proportion of problems with >2x speedup, and the overall speedup. However, the greatest improvement comes from using three agents; using more generates diminishing returns.
>
> **Increasing the interaction rounds.** We repeat the set up of Figure 3 from the paper by allowing the agents to interact longer. The results are summarized in the following table.
>
> | Round | Speedup |
> |------:|----------------:|
> | 1 | 1.90 |
> | 2 | 1.95 |
> | 3 | 2.01 |
> | 4 | 2.05 |
> | 5 | 2.12 |
> | 6 | 2.13 |
> | 7 | 2.13 |
> | 8 | 2.21 |
> | 9 | 2.22 |
> | 10 | 2.22 |
>
> Like in the previous case, the speedup improves when more computational resources are spent. However, if one visualizes the speedup curve, one will observe that there is a point after which the speedup nearly plateaus. This happens around round 8, which goes beyond the five rounds experimented in the paper submission.
>
> **Comparison with reasoning models.** We repeat the set up of Table 1 of the paper and add two models, o3 and DeepSeek-R1-Distill-Qwen-14B (abbrevated as DeepseekR1-14B), both with reasoning enabled. The results are summarized in the following two tables.
>
> **ParEval**
>
> | Model              | Serial Correct | Serial > 2× | Serial Speedup | OpenMP Correct | OpenMP > 2× | OpenMP Speedup |
> |--------------------|---------------:|------------:|---------------:|---------------:|------------:|---------------:|
> | o3                 | 0.77 ± 0.02 | **0.23 ± 0.04** | **2.21 ± 0.16** | 0.72 ± 0.03 | 0.58 ± 0.03 | **3.55 ± 0.27** |
> | DeepseekR1-14B     | 0.56 ± 0.05 | 0.12 ± 0.05 | 1.51 ± 0.19 | 0.44 ± 0.03 | 0.31 ± 0.02 | 1.68 ± 0.07 |
> | LessonL            | **0.91 ± 0.02** | 0.21 ± 0.01 | 2.16 ± 0.11 | **0.86 ± 0.01** | **0.62 ± 0.02** | 3.46 ± 0.03 |
>
> **PolyBench**
>
> | Model              | Serial Correct | Serial > 2× | Serial Speedup | OpenMP Correct | OpenMP > 2× | OpenMP Speedup |
> |--------------------|---------------:|------------:|---------------:|---------------:|------------:|---------------:|
> | o3                 | 0.64 ± 0.02 | **0.28 ± 0.02** | **1.71 ± 0.21** | **0.72 ± 0.04** | 0.56 ± 0.02 | 3.15 ± 0.23 |
> | DeepseekR1-14B     | 0.18 ± 0.04 | 0.05 ± 0.04 | 1.08 ± 0.07 | 0.30 ± 0.03 | 0.25 ± 0.04 | 1.68 ± 0.09 |
> | LessonL            | **0.77 ± 0.04** | 0.13 ± 0.04 | 1.32 ± 0.09 | 0.71 ± 0.03 | **0.62 ± 0.02** | **3.40 ± 0.12** |
>
> While DeepseekR1-14B does not deliver competitive results, to our surprise, o3 performs quite well. On both modes of ParEval and the serial mode of PolyBench, LessonL outperforms o3 regarding correctness, but it underperforms o3 regarding speedup. On the OpenMP mode of PolyBench, the observation flips.
>
> Note that the competitiveness of o3 comes at the cost of higher token consumption and consequently more money. o3 consumes a similar number of input tokens as GPT-4o but three times the number of output tokens. Under the current pricing, o3 will cost `$`0.830 to run a ParEval experiment, while GPT-4o costs `$`0.330 and our method costs only `$`0.326 (see Table 6 of the paper).
>
> ### W2. Lesson Quality and Generalization: The paper lacks mechanisms to verify lesson correctness beyond empirical performance, creating risk of error propagation through incorrect lessons. There's no discussion of handling conflicting or contradictory lessons. The paper doesn't evaluate cross-benchmark transfer—whether lessons from one benchmark help on another.
>
> Lesson verification is indeed an important and inherently challenging topic.
>
> While we do not have a good answer to how to perform reliable verification automatically (other than some non-robust ideas such as LLM judge or LLM debate), there are a few built-in mechanisms, due to the nature of LLMs or of our method, which mitigate error propagation. First, if a lesson is incorrect and it leads to slowdown or even incorrect code, it will be deprioritized by our effectiveness adjustment step, reducing the likelihood of its reuse. Second, our lesson solicitation process explicitly requires each lesson to begin with a clear statement of the code’s effectiveness, grounded in accurate execution feedback. This contrasts with methods that rely on self-reflection alone, which are more prone to errors.
>
> Regarding cross-benchmark transfer, our method focuses on using lessons to solve the same problem, rather than accumulating lessons over problems or even reusing lessons from different tasks. Building a knowledge-base of lessons is an important future direction.
>
>
> ### W3. Analysis: Missing analysis of which types of problems benefit most from this approach.
>
> We perform an analysis on the problem categories. The following table lists the speedup achieved by the best agent (among Deepseek7B, Qwen7B, and Qwen14B) for the corresponding category versus that by our multi-agent approach (LessonL).
>
> | Category   | Best single agent | LessonL | Relative improvement |
> |------------|--------:|---------:|-:|
> | dense_la   | 1.02 | **1.19** | 16.67% |
> | fft        | 1.02 | **2.23** | 118.63% |
> | geometry   | 8.09 | **9.80** | 21.14% |
> | graph      | 1.00 | **1.04** | 4.00% |
> | histogram  | 1.57 | **1.67** | 6.37% |
> | reduce     | **2.21** | 2.03 | -8.14% |
> | scan       | **5.95** | 5.91 | -0.67% |
> | search     | **1.07** | 1.02 | -4.67% |
> | sort       | 2.35 | **2.69** | 14.47% |
> | sparse_la  | **1.70** | 1.61 | -5.29% |
> | stencil    | 1.12 | **3.23** | 188.39% |
> | transform  | 1.01 | **1.03** | 1.98% |
>
> The table suggests that LessonL improves the speedup on many categories. Among them, "stencil" and "fft" enjoy the most significant improvement. For categories that LessonL suffers from degradation of speedup, the degradation is minor. In theory, LessonL is no worse than the best single agent. The degradation reflects randomness in LLM generation.
>
> ### Q1: How does performance scale with 5+ agents?
> See the response to **W1**.
>
> ### Q2: Is there a point of diminishing returns or negative interference?
> See the response to **W1**.
>
> ### Q3:  How do you handle potentially incorrect lessons? Have you observed cases where misleading lessons degraded performance across multiple agents?
> See the response to **W2**.
>
> ### Q4: Have you observed cases where misleading lessons degraded performance across multiple agents? (lessons that leads to slow down in the next iteration)
>
> It is rare that the agents produce worse programs over the rounds, but we do find one example from ParEval. This example is problem 13, finding the closest pair of points. There is a well-known, divide-and-conquer algorithm that runs in $O(n \log n)$ time (see [1, Chapter 33.4]). In the initial round, Deepseek7B produces a version that first sorts the points according to the x coordinate and then uses a nested for-loop to find the closest pair. Inside the inner loop, something clever is done by searching along a strip to avoid wasting unnecessary distance calculations. The worst-case complexity of this algorithm is $O(n^2)$.
>
> It turns out that this code runs the fastest, even faster than the well-known $O(n \log n)$ algorithm that was implemented by another agent in the initial round. The fact that the $O(n^2)$ algorithm runs the fastest is probably because of the data distribution that avoids hitting the worst-case complexity.
>
> Therefore, the success of both algorithms is summarized into lessons and passed on to the next round. In the next round, the agents pick the $O(n \log n)$ algorithm to implement, all bypassing the faster $O(n^2)$ algorithm.
>
> The spirit of this example is that even though the lessons all correctly describe the reason of speedup, the LLM agents should still be able to discern which one is the most helpful. This is too hard in this case (even for expert programmers, who would realize $O(n^2)$ is faster than $O(n\log n)$ in the first place?).
>
> [1] Corman et al. Introduction to Algorithm. 3rd Edition. The MIT Press. 2009.
>
> ### Q5: How does LessonL compare with recent test-time compute scaling approaches like those used in DeepSeek-R1(-Distill)
>
> See the response to **W1**.
>
> Thank you again for your constructive feedback. We believe that addressing these questions greatly improves the paper.

---

### Official Review · Reviewer_bGqu · 2025-07-06

**Clarity:** 4
**Significance:** 2
**Originality:** 2
**Rating:** 4
**Confidence:** 5

**Summary:**

This work proposed to develop a team of agents that can learn from each other's successes and failures so as to improve their own performance. Lessons are stored in the memory and passed on to other agents in the collective solution process.

This is an interesting work. The idea is straightforward and effective in terms of self-learning speed and performance improvement, when compared with other prompting / memory / reflection based work.

**Questions:**

See above

**Ethical Concerns:**

["NO or VERY MINOR ethics concerns only"]

**Limitations:**

See above

**Quality:**

3

**Strengths And Weaknesses:**

The paper is well written. I have no problem grasping the core idea of implementation and experiments in short period of time. However, the differentiation of this work with several previous works are not very clear. For example, for coding agents, there seem to be a lot of works last year developing multi-agent frameworks, such as https://arxiv.org/abs/2408.07060. They also have self-reflection or LLM judge steps to for better retrial.

1. Can the authors discuss the differentiations of this current against previous multi-agent frameworks with memory?
2. Can the authors clarify more about the lessons bank? How is this memory module designed and its implementation differentiation from RAG-based memory?

---

> ### Author Rebuttal · Authors · 2025-07-31
>
> Thank you for confirming that the paper is interesting and well-written.
>
> ### W1: Can the authors discuss the differentiations of this current against previous multi-agent frameworks with memory?
>
> A unique angle of our design is the lesson mechanism for agents to learn from each other's successes and failures. The lesson bank shares similarities with memory in LLM-powered agents. In what follows, we answer your questions regarding the differentiations of our design from memory and RAG-based memory.
>
> The memory of an agent typically includes the historical information during its task execution or problem solving [1]. The survey [2] further categorizes memory into short-term memory, long-term memory, and knowledge-retrieval as memory, the last of which extends internal history to external knowledge. Our lesson bank is the most aligned with long-term memory [3,4], since it deposits lessons accumulated throughout the entire problem-solving process.
>
> However, less obvious is how memory is organized and used when there are multiple agents working together. A straightforward design is that each agent maintains its own memory. In contrast, we design a *shared memory*, which facilitates agents to learn from each other. Without sharing, an agent would only learn from its own reflection or its past failures, but not the successes of others. Learning the lessons of other agents is especially important to gain new perspectives regarding how the code can be made even faster.
>
> It was brought to our attention by another reviewer that Agent Hospital [5] also proposes a shared memory (called "experience" by the paper), but the experiences are cumulated over many cases (patient treatment cases), while our bank of lessons is per problem and is not cumulated over problems. It remains interesting to see if cumulation of lessons can help, in which case we anticipate the lesson selection mechanism will need to be redesigned.
>
>
> In DEI [8], multiple agents propose solutions and then their solutions are evaluated by a DEI committee which determines the best one. In our system, however, we proceed in a multi-round fashion. In each round agents propose a solution and then generate "lesson" that may inspire other agents to improve.
>
> ### W2: Can the authors clarify more about the lessons bank? How is this memory module designed and its implementation differentiation from RAG-based memory?
>
> Regarding RAG-based memory, our approach is rather different from RAG [6,7]. If one were to use RAG to select lessons from the bank, one might use the current version of the code as the query and find the closest lesson from the bank. The closest match would be just the lesson generated for this code. In this case, the agent would only learn from its experience (and the most recent one, like a short-term memory!); it would not learn from its past or others' lessons. Instead, our design allows each agent to use lessons that describe how to achieve high speedups. Such lessons (1) may come from any time step and any agent; (2) may be ranked differently when taking into account the effectiveness adjustment; and (3) are not retrieved based on matching the current code.
>
> Thank you again for your constructive comments and questions.
>
> ### References
>
> [1] Zhang et al. A Survey on the Memory Mechanism of Large Language Model based Agents. Preprint arXiv:2404.13501, 2024.
>
> [2] Luo et al. Large Language Model Agent: A Survey on Methodology, Applications and Challenges. Preprint arXiv:2503.21460, 2025.
>
> [3] Wang et al. Voyager: An Open-Ended Embodied Agent with Large Language Models. Preprint arXiv:2305.16291, 2023.
>
> [4] Shinn et al. Reflexion: Language Agents with Verbal Reinforcement Learning. NeurIPS, 2023.
>
> [5] Li et al. Agent hospital: A simulacrum of hospital with evolvable medical agents. Preprint arXiv:2405.02957, 2024.
>
> [6] Lewis et al. Retrieval-Augmented Generation for Knowledge-Intensive NLP Tasks. NeurIPS, 2020.
>
> [7] Edge et al. From Local to Global: A Graph RAG Approach to Query-Focused Summarization. Preprint arXiv:2404.16130, 2024.
>
> [8] Zhang, Kexun, et al. "Diversity empowers intelligence: Integrating expertise of software engineering agents." arXiv preprint arXiv:2408.07060 (2024).

---

### Comment · Area_Chair_zWem · 2025-08-03
**Engage in Author-Reviewer Discussions**

Dear Reviewers,

Thanks for your efforts. Since authors have replied to the reviews, please check whether the rebuttal solves your concerns and respond to authors.

Best regards,

AC

---

### Decision · Program_Chairs · 2025-09-17

**Decision:**

Accept (poster)

**Comment:**

Summary:

This paper introduces a multi-agent framework called LessonL, which enables code LLMs to collaboratively solve coding problems through shared learning. The key contribution falls into the "lesson" mechanism. The lesson means concise knowledge pieces that explain why solutions succeed or fail, allowing agents to learn from each other's experiences. The LessonL framework operates through four iterative stages: (1) Lesson Solicitation from solution outcomes, (2) Lesson Banking in a central repository, (3) Lesson Selection based on speedup and semantic relevance, and (4) Solution Generation using selected lessons. The authors demonstrate that teams of small models (i.e., 7B-14B parameters) can outperform larger models like GPT-4o on both code optimization and generation tasks across six benchmarks.

Strengths:

1. The lesson-based collaboration mechanism is novel and inspiring compared with fixed role-based or voting-based multi-agent systems.

2. The experiments across six benchmarks are extensive and comprehensive. The empirical analysis does provide valuable insights.

3. This paper is overall well-written with clear demonstration of the proposed method and experimental settings.

Weaknesses:

1. The discussion on the scalability to more agents and rounds should be added.

2. The design principle of the proposed method is challenged by several reviewers (such as the heuristic lesson selection mechanism and the selection of specific LLMs), which needs more clarifications.

The author-reviewer discussion has solved most of the concerns. Thus, I highly recommend the authors to incorporate all the contents in the discussion to further improve the paper quality.